# OPENCUA: Open Foundations for Computer-Use Agents

**Xinyuan Wang**[*x] **Bowen Wang**[*x] **Dunjie Lu**[*x] **Junlin Yang**[*x] **Tianbao Xie**[*x] **Junli Wang**[*x]

**Jiaqi Deng**[x] **Xiaole Guo**[x] **Yiheng Xu**[x] **Chen Henry Wu**[c] **Zhennan Shen**[x] **Zhuokai Li**[x] **Ryan Li**[x] **Xiaochuan Li**[x]

**Junda Chen**[x] **Boyuan Zheng**[x] **Peihang Li**[x] **Fangyu Lei**[x] **Ruisheng Cao**[x] **Yeqiao Fu**[x] **Dongchan Shin**[x] **Martin Shin**[x]

**Jiarui Hu**[x] **Yuyan Wang**[x] **Jixuan Chen**[x] **Yuxiao Ye**[x] **Danyang Zhang**[x] **Yipu Wang**[m] **Heng Wang**[m]

**Diyi Yang**[s] **Victor Zhong**[w] **Y. Charles**[m] **Zhilin Yang**[m] **Tao Yu**[†x]

[x] XLANG Lab, The University of Hong Kong    [m] Moonshot AI
[s] Stanford University    [w] University of Waterloo    [c] Carnegie Mellon University

**Project Page:** https://opencua.xlang.ai

## Abstract

Vision-language models have demonstrated impressive capabilities as computer-use agents (CUAs) capable of automating diverse computer tasks. As their commercial potential grows, critical details of the most capable CUA systems remain closed. As these agents will increasingly mediate digital interactions and execute consequential decisions on our behalf, the research community needs access to open CUA frameworks to study their capabilities, limitations, and risks. To bridge this gap, we propose **OPENCUA**, a comprehensive open-source framework for scaling CUA data and foundation models. Our framework consists of: (1) an annotation infrastructure that seamlessly captures human computer-use demonstrations; (2) **AGENTNET**, the first large-scale computer-use task dataset spanning 3 operating systems and 200+ applications and websites; (3) a scalable pipeline that transforms demonstrations into state–action pairs with reflective long Chain-of-Thought reasoning that sustain robust performance gains as data scales. Our end-to-end agent models demonstrate strong performance across CUA benchmarks. In particular, **OPENCUA-72B** achieves an average success rate of **45.0%** on **OSWorld-Verified**, establishing a new state-of-the-art (SOTA) among open-source models. Further analysis confirms that our approach generalizes well across domains and benefits significantly from increased test-time computation. We release our annotation tool, datasets, code, and models to build open foundations for further CUA research: `https://opencua.xlang.ai`

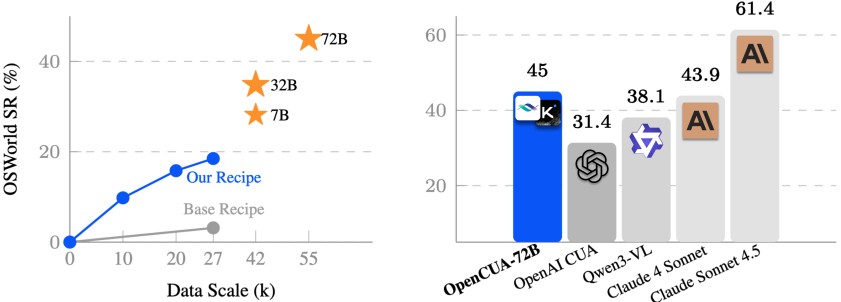

Figure 1: **OSWorld-Verified Performance** [44]: **Left**: OpenCUA recipe helps performance scaling with data size and model size. **Right**: OpenCUA-72B outperforms current open-source models and is comparable with Claude 4 Sonnet [2].

39th Conference on Neural Information Processing Systems (NeurIPS 2025).

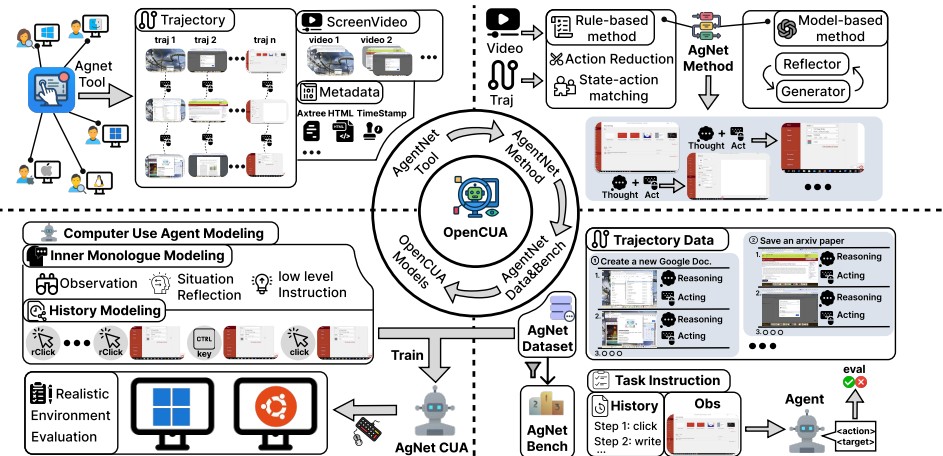

Figure 2: Overview of the OPENCUA framework. **Top left**: AGENTNET TOOL captures user interactions across OSes with screen videos and action flows. **Top right**: Raw demos are processed into state-action trajectories with reasoning and history. **Bottom right**: AGENTNET Dataset and Bench provide diverse tasks and offline evaluation with gold-standard actions. **Bottom left**: OPENCUA models are trained and able to execute in realistic environments.

# 1  Introduction

Computer-use agents (CUAs), powered by vision-language models (VLMs), aim to autonomously complete computer tasks and have great potential in facilitating daily and professional workflows. Despite their growing role in high-stakes decision-making, critical details including training data, architectures, and development processes about how state-of-the-art CUA systems are built remain closed and proprietary [1, 2, 4, 16, 29, 3]. As the lack of transparency limits technical advancements and raises safety concerns [32, 40, 10], the research community needs *truly open* CUA frameworks to study their capabilities, limitations, and risks.

However, current open-source attempts in CUA face significant challenges that impede progress. Firstly, there is no open-source scalable infrastructure for collecting diverse large-scale computer-use data - a complex requirement that involves the real-time capture of user interactions and state information, followed by transformation into agent-executable trajectories. Secondly, existing open-source graphical user interface (GUI) datasets remain limited in scope and scale due to the complexity and high cost of data collection; they either focus on specific domains (grounding [9, 43, 17, 46], mobile [31, 23], or web [13, 11]) or lack sufficient diversity for general computer-use applications. Furthermore, many CUA works provide insufficient details about their modeling strategies and training recipes, making replication difficult even with access to their collected data. These limitations collectively hinder advances in general-purpose CUAs and restrict a meaningful exploration of their scalability, generalizability, and potential learning approaches.

To address these challenges, we introduce OPENCUA, a fully open-source framework to scale the CUA data and the foundation models (Figure 2). To address infrastructure challenges, we first develop a user-friendly, cross-OS computer task annotation application AGENTNET TOOL that can be installed on personal computers to seamlessly record natural human demonstrations and corresponding computer states, without disrupting the user's workflow (Figure 2 top left). We then collect the AGENTNET dataset, including 22.6K open-domain computer task trajectories spanning over 100 applications and 200 websites across Windows, macOS, and Ubuntu (Figure 2 top right). This dataset authentically captures the complexity of human behaviors and environmental dynamics from users' personal computing environments. Furthermore, given that online CUA benchmarks such as OSWorld [44] require substantial environment setup effort and runtime resources, we curated AGENTNETBENCH based on our collected human demonstrations (Figure 2 bottom right). This offline benchmark provides multiple gold-standard actions per step, efficiently approximating online metrics to dramatically accelerate agent evaluation and development.

Critical to our OPENCUA framework is our (1) data processing pipeline and (2) novel modeling and training recipe for constructing CUA training data from human demonstrations. We first introduce an

Table 1: Comparison between AGENTNET and Other GUI Datasets

| Dataset | Tasks | Avg. Step | Env. Type | Personalized Env. | Human Traj. | Dom/ AxTree | Video | Inner Monologue |
|---|---|---|---|---|---|---|---|---|
| AndroidControl[20] | 15283 | 5.5 | Mobile | ✗ | ✓ | ✓ | ✗ | **Short** |
| AMEX[7] | 2991 | 11.9 | Mobile | ✗ | ✓ | ✗ | ✗ | ✗ |
| AitW[31] | 2346 | 8.1 | Mobile | ✗ | ✓ | ✓ | ✗ | ✗ |
| AitZ[54] | 1987 | 6.0 | Mobile | ✗ | ✓ | ✗ | ✗ | **Short** |
| GUI Odyssey[23] | 7735 | 15.3 | Mobile | ✗ | ✓ | ✗ | ✗ | ✗ |
| OS-Genesis[34] | 2451 | 6.4 | Mobile&Web | ✗ | ✗ | ✓ | ✗ | **Short** |
| WonderBread[39] | 598 | 8.4 | Web | ✗ | ✓ | ✓ | ✓ | ✗ |
| AgentTrek[48] | 10398 | 12.1 | Web | ✗ | ✗ | ✓ | ✓ | **Short** |
| Mind2Web[11] | 2350 | 7.3 | Web | ✗ | ✓ | ✓ | ✗ | ✗ |
| GUIAct[8] | 2482 | 6.7 | Web | ✗ | ✓ | ✓ | ✗ | ✗ |
| **AgentNet** | **22625** [1] | **18.6** | **Desktop** | ✓ | ✓ | ✓ | ✓ | **Long** |

action discretization pipeline that converts raw human demonstrations, which typically consist of videos and high-frequency, redundant keyboard/mouse actions, into state-action pairs feasible for vision language model training. Despite this, we observe that training on state-action pairs alone yields limited performance gains even as the dataset size scales (see Figure 1 Left). Our first key insight is that scaling agent capabilities requires augmenting these trajectories with reflective long Chain-of-Thought (CoT) reasoning. We propose a reflective CoT synthesis method that explicitly injects planning, memory, and reflection into the per-step reasoning process through natural language "inner monologue" (Section 3.1). Different from previous work, our reasoning traces are notably more detailed and contain reflection thoughts that help the agent detect and recover from errors. Moreover, we identify key modeling details that improve agent performance (Section 3.2), such as multi-image history. Finally, we show that carefully designing training data mixtures—including diverse reasoning and general text—is beneficial for computer-use agent training (Section 3.3).

Built upon our methodology, we developed strong computer-use agent models through supervised fine-tuning (SFT) (Figure 2 bottom left). Our results show that our approach enables robust performance scaling with increased data size (Section 4.2). Our model, OPENCUA-72B, achieves a success rate of **45.0%** (100 step) on **OSWorld-Verified** [44, 47], establishing a new state-of-the-art among the open-source models, even outperforming the proprietary OpenAI CUA (GPT-4o based) [29] and Claude 4 Sonnet [2]. It also has strong GUI grounding ability: **37.3%** (SOTA) on UI-Vision [25] and **60.8%** on ScreenSpot-Pro [19]. We did extensive experiments and analysis on various model structures and data scales in Section 5. Because of the diversity and coverage of our training data, our models show strong cross-domain generalization. Our agent models also show promising scalability with increased test-time compute, such as increased number of steps and larger $n$ in Pass@$n$ evaluation. We also did additional experiments, including grounding and robustness analysis. Finally, we also provide detailed ablations to justify the important design choices in our method and training recipe (Section 5). We open-source the complete suite of our OPENCUA framework, including the annotation tool, collected datasets, code, benchmark and models, providing open foundations for further CUA research.

## 2 AGENTNET Collection

OPENCUA aims to scale desktop computer-use data across diverse computer environments and user scenarios. We prioritize collecting demonstrations that follow natural user behavior, imposing the least additional constraints on how users interact with computers to improve the scalability of data collection. To this end, we developed **AGENTNET TOOL** and collected **AGENTNET** dataset, the first large-scale desktop agent task dataset. The introduction of AGENTNET TOOL and the data annotation strategies are demonstrated in the Appendix F.1

### 2.1 Task Definition

We model the agent's decision-making process – iterative observation of the computer state followed by action prediction – as a state-action transition trajectory: $(I, \langle s_0, a_0 \rangle, \langle s_1, a_1 \rangle, \ldots, \langle s_T, a_T \rangle)$. Given a task language instruction $I$ and initial state $s_0$, the agent sequentially predicts a action $a_i$ until goal state $s_t$ and performs the termination action $a_T$: $P(a_i | I, s_0, a_0, \ldots, s_i)$.

---

[1] A total of 41,428 trajectories were used to train OPENCUA-7B and OPENCUA-32B, and 27,804 trajectories were used for training OPENCUA-QWEN2-7B and OPENCUA-A3B. Moonshot AI annotated the Ubuntu subset and generously agreed to release 5K of the annotated trajectories to the public.

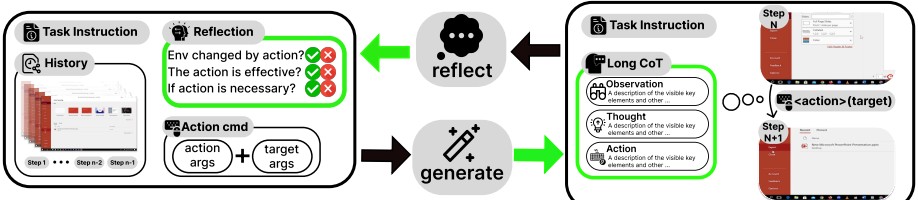

Figure 4: Reflective long CoT synthesis pipeline: generator and reflector iteratively generate and verify the reasoning components between the observation and ground-truth actions.

An important design choice in building computer-use agent is to convert compute state $s_i$ into model observation. In this work, we follow the recent trend of building pure vision-based computer agents [30, 50, 43] and use the screenshot of the computer as the observation for the agent. We use human computer-use actions, including keyboard and mouse movements, as the action space. To ensure the action space is applicable across various operating systems, we select a subset of PyAutoGUI actions and augment them with several necessary agent actions including the 'success' and 'fail' termination actions. The complete action space and its parameters are listed in Table 9.

## 2.2 AGENTNET Statistics

Our dataset consists of 22,625 human-annotated computer-use tasks, including 12K from Windows, 5K from macOS and 5K from Ubuntu, with screen resolutions ranging from 720p to 4K. Each trajectory averages 18.6 steps, reflecting the complexity of the task. As shown in Figure 3, the data spans over 140 applications and 190 websites, often involving multi-app workflows, professional tools, and uncommon features. Compared to previous GUI datasets (Table 1), AGENTNET is the first desktop trajectory-level dataset that is realistic, complex, diverse, and multimodal. The statistics are provided in the Appendix F.2.

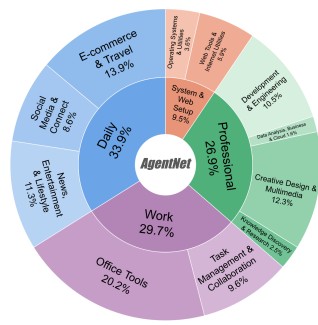

Figure 3: Domain distribution of tasks in AgentNet dataset

## 3 Training Computer-Use Agent Model

Our AGENTNET consist of task instructions $I$ and state-action $\langle s_i, a_i \rangle$ trajectories. However, we find that directly finetuning vision-language action (VLA) models on our 27K trajectories leads to poor performance (4.4% success rate on OSWorld [44], shown in Figure 1 Left "Base Recipe"). This section presents modeling and training recipe to enable scalable training of computer-use agent models, including novel reasoning augmentation, context encoding, and data mixtures techniques.

### 3.1 Synthesizing Reflective Long CoT Reasoning

Consistent with prior works [53, 50, 30], we find natural language reasoning crucial for generalizable computer-use foundation models, helping CUAs internalize cognitive capabilities. We propose a multi-stage CoT framework synthesizing structured reasoning per state-action pair $\langle s_i, a_i \rangle$. Inspired by Aguvis [50], our structured CoT includes three reasoning levels. The hierarchy begins with **L3**, contextual observation capturing salient visual and textual elements. Next, **L2** provides reflective reasoning analyzing state transitions, recalling previous steps, correcting errors, and planning subsequent actions. Finally, the model predicts **L1**, a concise executable action grounded in prior perception and thought. This L3→L2→L1 structure mirrors perceptual-to-agentic decision flow, equipping the model with coherent, interpretable reasoning trajectories that enhance generalization and robustness.

**Reflection augmentation for L2 reasoning** Incorrect or redundant annotations in human demonstrations are not all bad, as long as we can identify and use them to teach the identification and correction of model errors. Therefore, we designed a `reflector` to identify errors and generate reflection reasoning for each step. Our CoT synthesis framework extends the pipeline of Aguvis [50] and ActRe [52] by equipping the "Thought" with more comprehensive agent components, especially state-transit perception and reflection, and minimizing hallucination. As shown in Figure 4, our CoT synthesis pipeline consists of three components: `reflector`, `generator`, and `summarizer`.

The `reflector` inspects each step for correctness and redundancy by comparing screenshots before and after the action, examining the correctness of the action code itself and the generated CoT, expecially whether the "Action" aligns with the screenshot and code. When the step is incorrect or redundant, the `reflector` will elaborate reason and this step will be ignored during training. If the step is correct, the `reflector` will explain the differences the actions brings to the before and after state. The `generator` conditions on the full agent context—previous reflections, action history, task goal, screenshots, and action code—to generate structured CoT. To help the model ground coordinate-related actions more accurately, we incorporate *visual cues*: a red marker on the mouse action coordinate and a zoomed-in image patch (inspired by V* [41]). Finally, the `summarizer` refines vague user-written goals into more precise and aligned task objectives, and scores each trajectory for alignment, efficiency, and difficulty. Our method produces rich and meaningful CoTs that significantly improve model reasoning and planning. We use `claude-3-7-sonnet-20250219` as the base model for synthesizing the three components. The reflection helps agent model identify former errors and adjust future plan to make the task back to the right track. An example of error identification and correction in evaluation can be seen in Section G. Ablations in Section 5 demonstrate that this module is a important driver of performance gains.

## 3.2 Context Encoding and Test-Time Reasoning

For end-to-end agent models, history encoding plays a critical role in reasoning and planning.

**Context encoding**: (1) Textual history: We propose a structured inner monologue framework for textual history representation. Specifically, we adopt a dialogue-style history format representing the model's responses and corresponding computer screenshots. Following Aguvis [50], we use L1 CoT (Action) to represent previous steps because it is more token-efficient and allows longer history windows without truncation. Moreover, our inner monologue includes memory components, further compensating for the absence of richer CoT in earlier steps. See Section 5 for ablations on history representation. (2) Visual history: Multi-image screenshot history is essential for agent model performance because screenshots are lossless visual representation of history, providing more grounded context than textual summaries. However, including more images also increases input length and reduces training efficiency. By default, we use three screenshots as visual representation, as our experiments show that this achieves a balance between performance and efficiency (see Figure 9).

**Test-time reasoning format**: While the model is trained with a mixture of CoT levels, we adopt the L2 CoT format at inference time due to its richer reasoning content, which enhances the model's ability to reflect, plan, and reason. As shown in our ablation studies (Section 5), L2 CoT significantly improves test-time performance scalability—Pass@$n$ success rates on OSWorld increase markedly over Pass@1. In contrast, models lacking this reasoning augmentation exhibit limited scalability, highlighting the importance of strong reasoning signals at inference time.

## 3.3 Training Strategy

**CoT data mixture**: As we mentioned in the Section 3.1, our structured inner monologue contains three levels of CoT: L1 (Action), L2 (Thought + Action), and L3 (Observation + Thought + Action), each encoding complementary information for agent decision-making but has different conceptual information. L1 CoT has direct connection to the actual action; while there is helpful screenshot perception information in the L3 CoT, some irrelevant elements may also be described; L2 CoT contains planning and prediction that directly affect the predicted action in L1. Therefore, we propose to train the model with a mixture of all three levels of CoT to reinforce this different levels of connection. Data example of L1, L2, and L3 can be seen in Appendix H. We verify this design choice with ablations in Section 5. In our newest recipe of OPENCUA-72B, we also mix rollout trajectories in real environment in Appendix C.

**Mixture of grounding, planning, and general SFT data:** A general-purpose computer-use agent foundation model should be capable of both solving complex computer-use tasks and performing general reasoning grounded in world knowledge. To achieve this, we train on a mixture of data types that span both computer-use and general vision-language domains.

For grounding, we initialize the model using existing datasets such as ShowUI [21], UGround [14], and 189K bounding-box samples parsed from collected AXTree structures. For planning and reasoning, we include a diverse mix of Ubuntu and Windows/macOS demonstrations as well as

task-instruction-augmented samples (Section 3.1). To enhance generalization and reasoning ability, we additionally incorporate general supervised fine-tuning (SFT) data from the Kimi Team [35]. The general text data covering domains such as instruction following, mathematical reasoning, and long-context understanding. The general vision data includes domains such as OCR and vision QA data. This mixture ensures both GUI grounding and high-level reasoning capabilities across domains. Although these general data are not directly related to GUI environments, we find that mixing them improves the model's overall agentic performance. We present ablation results on this effect in Section 5.

**CUA training strategies:** Different CUA training strategies are introduced in Appendix C.

## 4 Experiments

### 4.1 Experimental Setup

**Models** Our experiments are conducted on multiple open-sourced vision-language models: Kimi-VL-A3B [35], Qwen2-VL-7B-Instruct [38], Qwen2.5-VL-7B-Instruct [5], Qwen2.5-VL-32B-Instruct [5] and Qwen2.5-VL-72B-Instruct [5][2]. Kimi-VL-A3B adopts a Mixture-of-Experts (MoE) architecture with 16B total parameters and 3B active during training and inference. It demonstrates moderate capability as a computer-use agent, including grounding and planning. Qwen2-VL and Qwen2.5-VL are general-purpose vision-language models, with Qwen2.5-VL exhibiting enhanced digital agent capabilities and expertise in high-resolution understanding. We conduct supervised fine-tuning (SFT) on these models and obtain our OpenCUA model variants: OPENCUA-A3B, OPENCUA-QWEN2-7B, OPENCUA-7B, OPENCUA-32B, and OPENCUA-72B.

**Evaluation** We evaluated our models on online evaluation benchmarks, offline agent evaluation benchmark, and GUI grounding benchmarks.

1. **Online agent evaluation:** (1) **OSWorld-Verified**: OSWorld [44] originally curated 369 human-crafted tasks covering a wide range of applications, each accompanied by its own environment setup and evaluation script. The OSWorld team has now verified every task, fixing those that were infeasible to test because of outdated dependencies, evaluation errors, or ambiguous instructions, and has released the improved benchmark as OSWorld-Verified [47][3]. Our results are obtained through the public evaluation by the OSWorld Team on AWS infrastructure. The results are reported in Table 2. (2) **WindowsAgentArena (WAA)** [6] contains 154 Windows-centric tasks, spanning native Windows applications as well as several open-source programs also featured in OSWorld. It can reflect the agent's online performance on Windows system.

   **Evaluation setting**: The resolution of the systems is $1920 \times 1080$. 4 clock tasks in WAA are dropped due to API and system image limitation. We adopt the L2 CoT format (Thought + Action) for all models, following results of our ablation in Section 5. Temperature is set to 0 for deterministic decoding during evaluation. The results in OSWorld-Verified are the average of 3 runs.

2. **Offline agent evaluation:** AGENTNETBENCH includes 100 representative held-out tasks covering diverse domains on Windows and macOS. The introduction and details of AGENTNETBENCH are in Appendix B. We also validated its correlation with online benchmark results in Figure 8.

3. **GUI grounding evaluation:** We evaluate our model's GUI grounding ability, the ability to map natural language instructions to specific actions within graphical user interfaces on 5 benchmarks: OSWorld-G [46], Screenspot-V2 [42], Screenspot-Pro [19], and UI-Vision [25]. OSWorld-G has 564 samples that systematically cover text matching, element recognition, layout understanding and fine-grained manipulation, with annotations for the element types required to solve each task. Screenspot-V2 includes screenshots from three platforms: mobile, desktop, and web. Screenspot-Pro focuses on high-resolution desktop environments, especially in professional settings. UI-Vision includes fine-to-course grained tasks to evaluate model's performance on understanding professional software, spatial reasoning, and complex actions in desktop environments.

---

[2]To align with the training infrastructure of the Kimi Team, we adopt the same chat template and tokenizer as Kimi-VL-A3B. M-RoPE in Qwen models is not implemented; we use 1D RoPE [33] instead.

[3]OSWorld-Verified Leaderboard: `https://os-world.github.io/` OSWorld-Verified blog: `https://xlang.ai/blog/osworld-verified`

## 4.2 Main Results

Table 2: **OSWorld-Verified results.** OPENCUA-72B achieves the best performance among all open-source models with an average success rate of **45.0%**, outperforming prior baselines by large margins. It also closes the gap to proprietary agents. This demonstrates the scalability and strength of our OpenCUA training pipeline. The details of each run and the Pass@3 evaluation results are shown in Appendix Table 7)

| Model | 15 Steps | 50 Steps | 100 Steps |
|---|---|---|---|
| **Proprietary** | | | |
| OpenAI CUA [29] | 26.0 | 31.3 | 31.4 |
| Seed1.5-VL [16] | 27.9 | - | 34.1 |
| Claude 4 Sonnet [2] | 31.2 | 43.9 | 41.5 |
| Claude Sonnet 4.5 [3] | - | - | 61,4 |
| **Open-Source** | | | |
| Qwen2.5-VL-32B-Instruct [5] | 3.0 | - | 3.9 |
| Qwen2.5-VL-72B-Instruct [5] | 4.4 | - | 5.0 |
| Kimi-VL-A3B [35] | 9.7 | - | 10.3 |
| UI-TARS-72B-DPO [30] | 24.0 | 25.8 | 27.1 |
| Qwen3-VL [5] | - | - | 38.1 |
| OpenCUA-7B (Ours) | $24.3^{+1.9}_{-1.3}$ | $28.1^{+0.7}_{-0.4}$ | $26.6^{+0.6}_{-0.5}$ |
| OpenCUA-32B (Ours) | $29.7^{+0.8}_{-1.5}$ | $34.1^{+1.0}_{-0.6}$ | $34.8^{+0.9}_{-1.0}$ |
| OpenCUA-72B (Ours) | **39.0** | **44.9** | $\mathbf{45.0}^{+1.1}_{-1.2}$ |

**Online agent evaluation.** Table 2 lists success rates of end-to-end agents for 15-, 50- and 100-step budgets on **OSWorld-Verified**. The details of each run, Pass@3 evaluation success rate, and the results of OPENCUA-A3B and OPENCUA-7B are demonstrated in Appendix Table 7. Besides quantitative metrics, we also provide a qualitative case study in Appendix G, which shows that our agent can recognize and correct earlier mistakes during long-horizon execution, ultimately enabling more reliable task completion.

1. **Proprietary models still lead, but the gap is closing.** Claude Sonnet 4.5 achieves 61.4% at 100 steps, followed by Claude 4 Sonnet at 43.9%. Our OPENCUA-72B reaches 45.0% in 100 steps, establishing a new state-of-the-art among open-source systems, closing the gap from open-source models to Claude models.

2. **OpenCUA method applies to models with different architectures and sizes.** We apply our method on 5 model architectures, including Kimi-VL-A3B [35], Qwen2-VL-7B [38], Qwen2.5-VL-7B [5], Qwen2.5-VL-32B [5] and Qwen2.5-VL-72B [5], covering MoE and dense structures and activation parameter number from 3B to 72B. As the results in Appendix 7, all the models' performance improves by a large margin upon the base models. In confirming that our data pipeline and training recipe scale effectively with model capacity. Furthermore, their performance scales with the model size - OPENCUA-72B consistently outperforms OPENCUA-7B under every step budget.

3. **Effect of step limit.** Most agents profit from a larger step budget from 15 to 50 steps, yet the gain from 50 to 100 steps is often smaller than that from 15 to 50 steps.
   - Claude 4 Sonnet: +12.7% from 15 → 50 steps, then − 2.4% from 50 → 100.
   - OPENCUA-32B: +4.4% (29.7 → 34.1) from 15 → 50 steps, and +0.7% (34.1 → 34.8) from 50 → 100 steps.
   - Base models including Qwen2.5-VL and Kimi-VL: marginal gains (< 1%), reflecting limited step scale capability of base models.

   Even top-tier agents (e.g., Claude 4 Sonnet) still struggle to benefit significantly on larger step budgets for truly long-horizon tasks. The modest gain when moving from a 50-step to a 100-step limit stems from two main factors: (i) most tasks need more than 15 but fewer than 50 GUI actions, so the extra head-room often goes unused; (ii) current models are still not good at recognising their own mistakes, recovering from errors, and deciding when to stop—hallucinations and repetitive loops frequently waste the additional steps. We did a detailed error study in Apendix D.4.

Table 3: Computer-use agent performance on AGENTNETBENCH. Coord actions: *click, rightClick, doubleClick, moveTo, dragTo, scroll*; Content actions: *write, press, hotkey*; Function action: *terminate*.

| Model | Coord. SR | Content SR | Func. SR | Avg. SR |
|---|---|---|---|---|
| Qwen2.5-VL-7B [5] | 50.7 | 40.8 | 3.1 | 48.0 |
| Aguvis-7B [50] | 56.7 | 43.3 | 0.0 | 52.4 |
| Qwen2.5-VL-32B [5] | 66.6 | 47.2 | 41.5 | 64.8 |
| Qwen2.5-VL-72B [5] | 67.2 | 52.6 | 50.5 | 67.0 |
| OpenAI CUA [29] | 71.7 | 57.3 | **80.0** | 73.1 |
| OpenCUA-7B (Ours) | 79.0 | 62.0 | 44.3 | 75.2 |
| OpenCUA-32B (Ours) | **81.9** | **66.1** | 55.7 | **79.1** |

4. **OpenCUA models achieve markedly higher Pass@$n$ scores.** As reported in Appendix 7, the Pass@3 success rate of OPENCUA-32B on OSWORLD-Verified jumps from 34.2%(Pass@1) to **45.6%**. The performance of OPENCUA-72B increases from 45.0%(Pass@1) to **53.2%**(Pass@3).This large margin suggests ample headroom for future post-training, reranking or multi-agent methods. Additional analyses of test-time upper bounds and robustness are provided in Section 5.

These observations validate the effectiveness of our pipeline and highlight the remaining headroom for open-source agents to close the gap with larger proprietary models.

**Offline benchmark evaluation.** AGENTNETBENCH is constructed from representative tasks in the OPENCUA dataset. It is a held-out testset including 100 task trajectories on Windows and macOS. To account for domain similarity, we group models into *Zero-shot models*—those not trained on AGENTNET (Qwen2.5-VL-7B/32B/72B, Aguvis-7B, and OpenAI CUA (GPT-4o))—and *Fine-tuned models*, namely our OPENCUA-7B and OPENCUA-32B. As summarized in Table 3:

1. **Scaling with model size.** In the zero-shot group, performance scales with model size; the specialist Aguvis-7B surpasses the general-purpose Qwen2.5-VL-7B.

2. **OpenAI CUA generalize well on unseen tasks.** OpenAI CUA outperforms all open-source zero-shot models and approaches fine-tuned agents, particularly excelling in terminate-state detection and content-based actions.

3. **Offline benchmark has correlation with the ranking on online benchmark.** The overall ordering generally matches the online leaderboard in Table 2: OPENCUA-32B > OpenAI CUA > Qwen2.5-VL models. OPENCUA-32B and OPENCUA-7B are relatively higher due to the alignment of the domain and action space.

4. **Coordinate-action performance reflects model grounding performance.** OPENCUA-32B exceeds OPENCUA-7B on coordinate-based actions, aligned the grounding results in Table 5.

**Performance scaling with data scaling.** We explore the effect of data scale on Qwen2-VL from three perspectives: cross-domain data, in-domain data, and out-of-domain data. We first investigate cross-domain data in Figure 1. Specifically, we compare three training settings: (1) 7K Ubuntu data, (2) 7K Ubuntu + 14K Win&Mac data, and (3) 10K Ubuntu + 17K Win&Mac data. On OSWorld, performance improves significantly from **9.8%** to **18.5%**, despite the added Win&Mac data coming from a different platform. This indicates that even out-of-domain data can substantially enhance generalization and reasoning ability, rather than causing negative transfer. To further study the impact of in-domain and out-of-domain data scale, we randomly sampled 3K, 7K, 10K trajectories from Ubuntu data and 3K, 7K 14K from Win&Mac.

As shown in Figure 5, performance scales consistently across all benchmarks with both in-domain and out-of-domain data. When increasing the Ubuntu data from 3K to 10K, the average performance improves by 72%. Scaling the Win/Mac data from 3K to 14K yields a 125% improvement on average. These results demonstrate a strong positive correlation between data quantity and agent performance, highlighting the importance of large-scale, diverse CUA data for model generalization.

**Performance on grounding benchmarks.** The evaluation results on grounding benchmarks are demonstrated in the Appendix D.2.

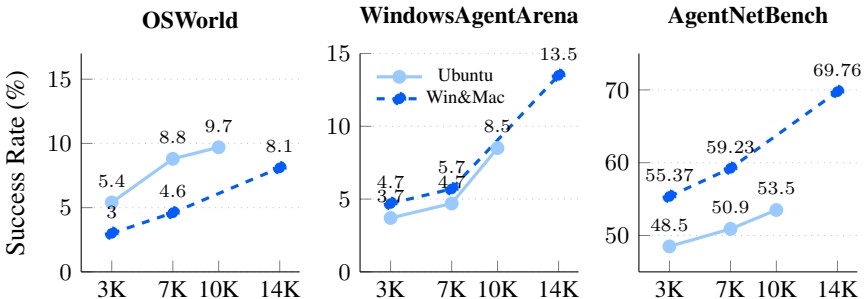

Figure 5: Scaling curves on three benchmarks as data volume from various OS domains increases.

## 5 Analysis

**Model performance upperbound analysis by scaling test-time compute** We further explore our model's performance upperbound by doing Pass@$n$ evaluation on OSWorld. We set the temperature to 0.1 and evaluate OPENCUA-QWEN2-7B for 16 times on the budget of 15, 30 and 50 steps and calculated the pass@1,4,8,16 success rate. In Table 6, we find: (1) There is a significant performance gap of our model between Pass@1 and Pass@16. On 15 step, the success rate increases from 16.9 to 34.6 (+104%), while on 50 step, the increacement is even large from 18.4 to 39.2 (+113%). (2) With larger $n$, the performance gains from increasing the step budget become more significant. (3) Online benchmarks have a large variance. To study model robustness, we did Pass@n evaluation when temperature is 0 in Figure D.5 and find higher temperature leads to higher Pass@n performance but lower Pass@1 performance.

We then investigated in the results and find variance comes from these factors:

1. The agent chooses different solutions in different runs. For example, in the task "Re-open the last closed browser tab," the agent sometimes uses Ctrl + Shift + T (only one step) and other times navigates through the history menu (needs many steps). Agents may fail on harder solutions.

2. Minor omissions or extras. In Chrome or VSCode settings, forgetting to click "Save" (or performing an additional stray click) converts a correct solution into failure.

3. Environment dynamics: Occasional CAPTCHA dialogs, machine variability, and network latency can change the interaction sequence and lead to inconsistent outcomes.

**Cross-platform training improves generalization, even with domain differences.** As shown in Figure 5, there is a consistent performance gap between models trained on different domains. Models trained on Ubuntu data perform better on OSWorld, while those trained on Windows/macOS data perform better on WindowsAgentArena and AGENTNETBENCH. This domain gap reflects the underlying differences in GUI layouts, system styles, and application behavior across platforms. OSWorld primarily focuses on applications and websites aligned with Ubuntu environments, whereas WindowsAgentArena contains several OSWorld Windows-specific applications. Interestingly, the performance gap between training on Win&Mac data versus Ubuntu data is narrower on WAA than on OS-

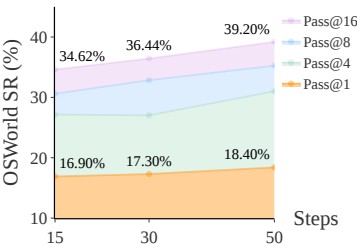

Figure 6: OSWorld Pass@N performance curves of OPENCUA-QWEN2-7B, temperature=0.1

World, suggesting that application-level knowledge can partially transfer across operating systems, even if interface styles differ.

**Other analysis and ablation studies.** More analysis on model robustness, error analysis and the ablation studies of reasoning format, history image number, CoT mixture, general data, and reflective long CoT are elaborated in the Appendix D.3

# 6 Related Work

**CUA benchmarks and datasets**  Autonomous computer-use agents are now judged primarily through *execution-level* benchmarks that embed the agent inside genuine software environments rather than synthetic simulators. On the desktop side, suites such as [45, 6, 55] orchestrate multi-step workflows that span office productivity, source-code editing, file management, and operating-system configuration across Linux, Windows, and macOS. For the web domain, campaigns including [56, 18, 11, 51, 12] deploy agents on self-hosted or live sites with dynamic content, long navigation chains, and non-trivial information-retrieval subtasks. To support training at the necessary scale, several high-volume data pipelines have appeared: tutorial-to-trajectory conversion for browser tasks [48, 22], cross-device grounding and action logs [17, 9, 15], plus our own collection of 27 K desktop demonstrations that pair screenshots, low-level mouse/keyboard events, and reflective chain-of-thought annotations.

**CUA frameworks and models**  Approaches to building computer-use agents can be grouped into three broad categories. First, text-based language models operate on structured GUI meta-data—such as DOM trees or accessibility labels—and issue symbolic commands; representative work ranges from early page-centric agents [24] to more recent language-only planners that still eschew raw pixels [49]. Second, vision-centric agents integrate screen imagery. Some focus on grounding—learning to associate natural-language references with bounding boxes or coordinate clicks [15, 42]—while others pursue end-to-end policies that translate full screenshots directly into action sequences [50, 30, 29, 1]. Third, modular agent frameworks wrap large language models with additional components—specialised vision encoders, hierarchical or search-based planners, episodic memory, and tool APIs—to tackle long-horizon tasks requiring perception, reasoning, and control [57].

# 7 Conclusion

We presented OPENCUA, a comprehensive open-source framework addressing critical gaps in computer-use agent development. By offering annotation infrastructure, data processing pipelines, diverse datasets, effective training recipes, and efficient evaluation benchmarks, we establish essential foundations for CUA research. Our models demonstrate strong performance across benchmarks while exhibiting clear data scaling laws and cross-domain generalization capabilities. By releasing all components—tools, datasets, code, and models—we aim to accelerate transparent CUA research, enabling the community to systematically investigate these agents' capabilities, limitations, and risks as they increasingly mediate our digital interactions and execute consequential decisions on our behalf.

# 8 Aknowledgement

The authors of this paper were supported by the ECS (27212023) and Areas of Excellence Scheme (AoE/E-601/24-N) from RGC of Hong Kong. We thank Yu Su, Caiming Xiong, and the anonymous reviewers for their insightful discussions and valuable feedback. We are grateful to Moonshot AI for providing training infrastructure and annotated data. We also sincerely appreciate Jin Zhang, Hao Yang, Zhengtao Wang, and Yanxu Chen from the Kimi Team for their strong infrastructure support and helpful guidance. The development of our tool is based on the open-source projects-DuckTrack [36] and OpenAdapt [27]. We are very grateful to their commitment to the open source community. Finally, we extend our deepest thanks to all annotators for their tremendous effort and contributions to this project.

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

# Table of Contents in Appendix

# A    Limitations

The scalability and quality of AGENTNET dataset is inherently limited by human annotation efforts. Although AGENTNET TOOL streamlines the data collection process, expanding the dataset beyond its current size would require additional human resources. Also, annotators may not have the expertise to complete computer tasks in the most effective approach (i.e. using shortcuts, even write coding scripes). Exploring alternative data sources or semi-automated annotation methods could help address this limitation. Additionally, though OPENCUA strives to collect authentic computer-use data from personal devices, our ethical requirement for explicit informed consent regarding data practices inevitably introduces selection bias. While our dataset maintains high diversity and authenticity, it necessarily excludes data from users who, upon understanding the potential risks, opt not to participate. This is a limitation we accept to uphold responsible data collection.

# B    AGENTNETBENCH

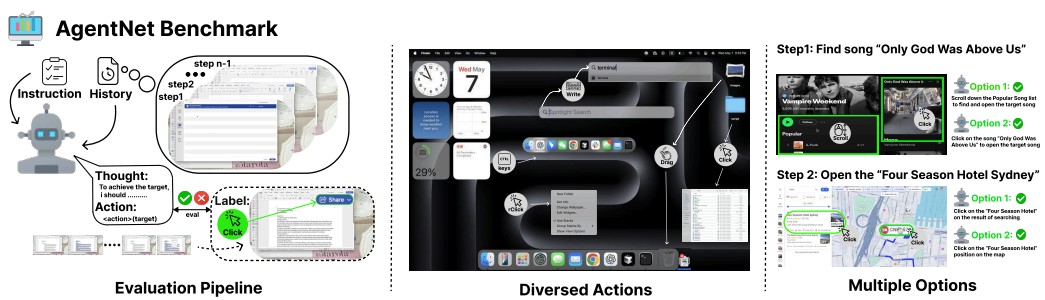

Figure 7: Illustration of the AgentNet Benchmark evaluation pipeline

There are several online benchmarks [44, 6] that evaluate agent performance in desktop environments. However, these online benchmarks typically require substantial computational resources for environment setup, making evaluations expensive, slow, and difficult to reproduce consistently through time due to their reliance on dynamic environments. Meanwhile, they only provide sparse, high-variance signals (i.e., trajectory-level accuracy). Another line of works, such as ComputerAgentArena [37], evaluate agent's performance of open-ended tasks on live environments through human preference. To address the limitations of online evaluation benchmarks, we introduce an offline CUA evaluation benchmark, AGENTNETBENCH, comprising 100 representative tasks selected from the AGENTNET dataset. Tasks were strategically chosen from the center of sub-domain clusters (as detailed in Section F.2.1), ensuring diversity and representativeness across applications and websites on Windows and macOS platforms. Each task was manually reviewed to refine goals and remove redundant actions. Notably, we manually provide multiple valid action options at each step because of the inherent multiplicity of valid actions in computer-use tasks.

**Benchmark statistics and evaluation dimensions**    The AGENTNETBENCH maintains a balanced domain distribution consisting of 38 Work tasks, 29 Daily tasks, 24 Professional tasks, and 9 System & Web Setup tasks. The tasks are split between two operating systems, with 61 tasks from Windows and 39 tasks from macOS. Screen resolutions are categorized into three levels (high, medium, and low) as detailed in Table 4 (note that, for practical purposes, all images in the benchmark are resized from their original resolutions). The distribution of actions within these tasks and additional benchmark statistics are also presented comprehensively in Table 4.

**Multiple action choices for enhanced accuracy**    Previous offline benchmarks [31, 20] typically define a single ground-truth action at each step. This practice can negatively impact accuracy by disregarding alternative valid choices that an agent may reasonably make in real-world interactions. In contrast, in AGENTNETBENCH, we annotate multiple plausible action choices for each step to better reflect real-world decision-making variability.

**Step success rate calculation and action matching criteria**    To calculate the Step Success Rate (Step SR), we evaluate the correctness of agent actions at each individual step using precise matching

Table 4: Comprehensive Statistics of AgentNetBench

| Domain Distribution | | Operating System Distribution | |
|---|---|---|---|
| Work | 38 | Windows | 61 |
| Daily | 29 | macOS (Darwin) | 39 |
| Professional | 24 | | |
| System & Web Setup | 9 | | |
| **Resolution Distribution** | | **Overall Statistics** | |
| High | 20 | Total Tasks | 100 |
| Medium | 33 | Avg. Steps/Task | 17.63 |
| Low | 47 | Total Actions | 2143 |
| **Action Distribution** | | | |
| click | 850 (67.0%) | doubleClick | 19 (1.5%) |
| rightClick | 17 (1.3%) | press | 28 (2.2%) |
| dragTo | 27 (2.1%) | write | 137 (10.8%) |
| moveTo | 45 (3.5%) | hotkey | 30 (2.3%) |
| scroll | 18 (1.4%) | terminate | 100 (7.6%) |

criteria tailored to different action types. For coordinate-based actions (*e.g.*, `click`, `doubleClick`, `moveTo`, `dragTo`, `rightClick`, and `hscroll`), we define bounding boxes around each action's target location; the agent earns the step success point if its predicted coordinates fall within these bounding boxes. For content- or keyboard-based actions, such as `write`, we measure correctness by computing the edit distance between the predicted and ground-truth text; actions like `hotkey` and `press` require perfect matches of the specified key combinations. For the `scroll` action, correctness depends on two key criteria: the agent's output coordinates must be within the designated bounding box, and the scrolling direction must exactly match the ground truth. Finally, the correctness of the `terminate` action depends on the agent appropriately terminating at precisely the correct step – neither prematurely nor delayed. Considering the distribution of actions (see Table 4), these fine-grained evaluation rules ensure accurate and fair evaluation of agent capabilities in diverse interaction scenarios.

**AGENTNETBENCH strongly correlates with online benchmark performance** The offline benchmark primarily assesses an agent's decision-making capability by evaluating its first-choice accuracy at each task step. While agents can leverage self-reflection to recover from errors made in earlier steps, offline and online SRs should correlate under a low step budget. Figure 8 and Table 3 indeed demonstrate a clear positive correlation, specifically following a power-law relation between the online task success rate (under a 15 step budget) and the offline step success rate. Therefore, metrics obtained from our offline benchmark provide a reliable indicator of an agent's foundational proficiency and its adaptability to realistic, resource-constrained online tasks.

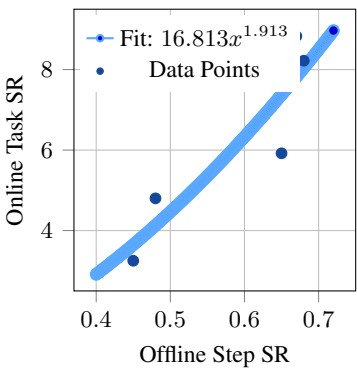

Figure 8: Offline vs. Online evaluation.

# C  CUA Training Strategies

Depending on computing budget, dataset availability, and the target model—e.g., building a specialized computer-use agent or a general-purpose VLM with agentic capabilities—different training strategies may be adopted. Following Aguvis [50], which proposes a two-stage curriculum training (Stage 1 for grounding and Stage 2 for planning and reasoning), we further explore three strategies:

1. **Stage 2 only:** When training resources are limited and the focus is on computer-use agent data, we aim to adapt a general open-source VLM into a specialized CUA. To preserve general instruction-following ability, we use a training mix of 70% CUA data (with a planning-to-grounding ratio of 4:1) and 30% general SFT data. We fine-tune Qwen2-VL with 30B tokens and Kimi-VL-A3B with 20B tokens. Both models exhibit strong improvements on CUA tasks (see Table 7).

2. **Stage 1 + Stage 2:** With more resources and diverse data, a staged approach yields better performance. In Stage 1, we enhance grounding and understanding using grounding trajectories, tutorial-style demonstrations, state-transition caption data, general vision language tasks, and general text SFT data. We trained Qwen2.5-VL-32B on this mixture for 40B tokens. In Stage 2, we shift focus to CUA planning, using 45% planning, 20% grounding, and the rest general data. This results in OPENCUA-32B, which achieves substantial gains in both grounding and planning benchmarks (Table 2, Table 5). Using the same strategy, we trained Qwen2.5-VL-72B but with more data to build our OPENCUA-72B. In addition to more annotated trajectories, we also used 8k trajectories rolled out in an Ubuntu environment using o3 [28]+Jedi [46]. We designed the CoT to emphasize the most effective information—especially reflection—and placed this data in Stage 2 to transform the model's CoT into a more efficient, information-dense format. In this way, the model first learns rich behaviors (reflection, planning, . . . ) in Stage 1 and then leverages them more effectively in Stage 2.

3. **Joint training:** To build a general-purpose VLM with strong CUA abilities, we adopt joint training across domains with balanced data mixing. Given the complexity of multi-image trajectory data, we train these samples for 3 epochs to ensure sufficient learning. Based on Qwen2.5-VL-7B, we train a model on 200B tokens budget, maintaining a data ratio of 20% planning, 20% grounding, and 60% general. The resulting model, OPENCUA-7B, achieves state-of-the-art performance among 7B-scale open-source CUAs, achieving 27.3% success rate on OSWorld Online Evaluation Platform.

# D  Experiment Setup, Extra Experiments and Analysis

## D.1  Experiment Setup and Training settings

All models are trained on the Kimi Team's infrastructure with the Megatron framework and DeepSpeed (ZeRO-3). We employ three training strategies:

1. **Stage-2 only.** OPENCUA-QWEN2-7B and OPENCUA-A3B share a configuration of sequence length 32,768, learning-rate $2 \times 10^{-5}$, weight-decay 0.1, and global batch size 384 (512 in ablations) on 96 × A100GPUs. They are trained on 18k Win&macOS + 10k Ubuntu trajectories. OPENCUA-QWEN2-7B runs for 3,400 steps (about 45 h) after a 400-step grounding warm-up; OPENCUA-A3B runs for 2,000 steps (about 10 h).

2. **Stage-1 + Stage-2.** OPENCUA-32B is first pretrained on 35B tokens of general text, vision, and grounding data (batch 3,584, LR $3 \times 10^{-5}$, 224 × A100). We take the step-1200 checkpoint. Stage-2 then continues for 60B tokens on trajectory + general + grounding data (batch 512, LR $2.5 \times 10^{-5}$, 128 × A100) using 18k Win&macOS + 20k Ubuntu trajectories. The final model corresponds to step 4,700. For OPENCUA-72B, we trained Qwen2.5-VL-72B using more data. Stage 1 contains 250B token data (batch size 600, LR $2.5 \times 10^{-5}$ decay to $1.5 \times 10^{-5}$, 480 × A100). Stage 2 contains 16B token (batch size, LR $1.5 \times 10^{-5}$ decay to $2 \times 10^{-6}$, 480 × A100).

3. **Joint training.** OPENCUA-7B is trained end-to-end on the full data mixture for 200B tokens (18k Win&macOS + 20k Ubuntu trajectories) with batch 512, peak LR $2.5 \times 10^{-5}$ (min LR $3 \times 10^{-6}$), decay tokens 200B, on 128 × A100 for eight days. The best checkpoint is at step 14,600.

Table 5: GUI grounding results.

| Model | OSWorld-G | Screenspot-Pro | Screenspot-V2 | UI-Vision |
|-------|-----------|----------------|---------------|-----------|
| Qwen2.5-VL-7B | 31.4 | 27.6 | 88.8 | 0.85 |
| Qwen2.5-VL-32B | 46.5 | 39.4 | 87.0 | – |
| UI-TARS-72B | 57.1 | 38.1 | 90.3 | 25.5 |
| OpenCUA-A3B (Ours) | 48.6 | 28.5 | 91.4 | – |
| OpenCUA-Qwen2-7B (Ours) | 45.7 | 23.7 | 88.5 | – |
| OpenCUA-7B (Ours) | 55.3 | 50.0 | 92.3 | 29.7 |
| OpenCUA-32B (Ours) | **59.6** | 55.3 | **93.4** | 33.3 |
| OpenCUA-72B (Ours) | 59.2 | **60.8** | 92.9 | **37.3** |

## D.2 GUI grounding evaluation.

Table 5 and Table 2 reveal several key findings:

1. **OPENCUA-72B and OPENCUA-32B rank the first.** OPENCUA-72B and OPENCUA-32B are the best-performing agentic model on all mainstream GUI-grounding benchmarks. OPENCUA-72B achieves **60.8**% on ScreenSpot-Pro and **37.3**% (SOTA) on UI-Vision. Their advantage stems from (i) the substantially larger grounding corpus used in Stage-1 training and (ii) its larger 32B parameter scale.

2. **Joint-training boosts OPENCUA-7B.** OPENCUA-7B also scores competitively because large-scale grounding data are injected during joint training.

3. **Pixel-budget advantage of Qwen2.5-VL.** Qwen2.5-VL based models uses a higher max-pixels limit (12,845,056 vs. 829,440 for Qwen2-VL and Kimi-VL-A3B), yielding stronger results on high-resolution ScreenSpot-Pro. OPENCUA-32B achieves 55.3% on Screenspot-Pro, and OPENCUA-7B achieves 50.0%.

4. **Grounding alone is not enough.** Although Qwen2.5-VL-32B matches or even surpasses OPENCUA-QWEN2-7B and OPENCUA-A3B on OSWorld-G and ScreenSpot-V2, OpenCUA models achieve far higher success rates on the full OSWorld benchmark (OPENCUA-A3B 19.9% and OPENCUA-QWEN2-7B 23.0% shown in Appendix 7). This shows that solid grounding is necessary yet insufficient for realistic agentic tasks in the environment; high-level planning and reflective reasoning ultimately drive reliable task completion.

5. **Domain correlation.** Because OSWorld-G [46] is collected in an Ubuntu environment, its scores correlate more closely with the online OSWorld evaluation than with ScreenSpot-Pro and Screenspot-V2.

## D.3 Ablation Studies

**L2 reasoning format achieves the best inference performance.** Note that we trained the models with mixed reasoning format (L1, L2, L3, see Section 3.1). In this part, we explore which type of thinking format is the best at inference. We test OPENCUA-QWEN2-7B and OPENCUA-A3B on OSWorld in 15 steps. As in Table 6, using the L2 format, the performance is higher than L1 and L3. This result is actually different from the conclusion from previous work [50, 30] that L1 outperforms L2. We think this is because our L2 CoT has higher quality (e.g., planning and reflection), which can help the model make better decisions. On the other hand, L3 underperforms L2. By case study, we find that when model describes the information in the screenshot, there tend to be many elements irrelevant to the task or the next action, which may mislead the model. In summary, our results show that the right choice of high-quality, non-redundant reasoning can leverage VLM's language reasoning capability to improve the agent performance.

**Using a moderate number of visual history images and concise textual history yields the best trade-off between performance and efficiency.** We ablate history representation from both visual and textual perspectives. For vision, we vary the number of history screenshots (1, 3, 5) and fine-tune Qwen2-VL-7B on 7K trajectories. As the OSWorld results shown in Figure 9, using multiple screenshots substantially improves performance over single-image inputs, as GUI agents rely entirely

Table 6: Ablation results on OSWorld for different Chain-of-Thought (CoT) settings.

| Ablation | CoT Variant | SR (%) |
|---|---|---|
| CoT Mixture | L2 | 13.1 |
| | Mixture-CoT | 18.5 |
| Reflective Long CoT | Short-CoT | 11.5 |
| | Advanced-CoT | 15.3 |
| Test-time Reasoning Format | L1 | 16.9 |
| | L2 | 18.5 |
| | L3 | 17.6 |

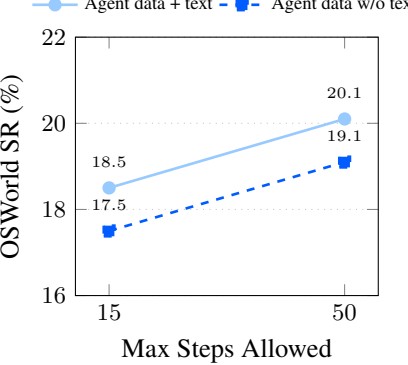

Figure 10: General text data ablation.

on vision for observing state changes. However, increasing from 3 to 5 images yields marginal gains while incurring 3K more context tokens and delayed convergence, suggesting diminishing returns.

On the textual side, we compare L1 and L2 history under the same 3-image setting. In Figure 9, L2 history offers no benefit and may introduce hallucinations that distract attention, while also reducing training efficiency. Hence, we adopt L1 CoT + 3 images as the default setting.

**Training with a mixture of CoT formats outperforms using only L2 reasoning.** Since our best performance is from L2 CoT inference, and L3 and L1 is lower than L2, we did an ablation of only training the L2 data instead of the mixture of L1, L2, and L3. We use the same recipe as our OPENCUA-QWEN2-7B, but only replace the mixture CoT data with L2 data. As the OSWorld result in Table 6, the model trained on L2 data using the same steps as OPENCUA-QWEN2-7B, but the performance drops to 13.1, which is aligned with the conclusion of Aguvis [50].

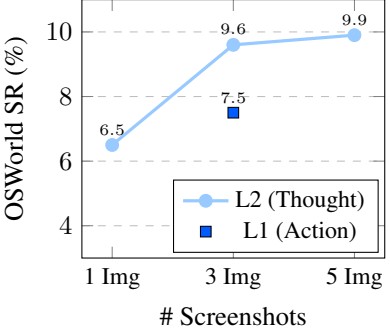

Figure 9: Effect of history representation: L1 (Action) benefits from more screenshots, while L2 (Thought) at 3 images lags behind.

**General-domain text data provides a positive effect to agent performance.** As we mentioned in Section 3.3, we used 35% general text data in our main experiment, so we also use the same agent data without the text data to fine-tune Qwen2-VL-7B with grounding warm-up stage for 2400 steps (approximately the same amount of agent data tokens) to ablate its influence. According to Figure 10, the general text data slightly improves model's agentic performance. Therefore, adding text data from totally different general domains doesn't impair the agent model's performance, on the contrary, helps improve the performance. We think the reason is that the general text data may help agent model's generalization and instruction understanding.

**Reflective long CoT significantly boosts performance by improving error correction.** To understand the effect of reflective long CoT (Secion 3.1), we do an ablation study on Qwen2-VL-7B with 14K Win&Mac and 3K Ubuntu trajectories. Without reflective long CoT, the CoT reduces to that used by Aguvis [50]. In Figure 11, we see that reflective long CoT improves the performance from 11.5 to 15.3. Since the reflective reasoning focuses on error correction, we conjecture that the improvement comes from improved self-correction capability.

### D.4 Error Study

After inspecting our online evaluation results, we group the failures into the following categories:

**1. Insufficient task knowledge** Foundation models may lack domain-specific GUI knowledge or the procedural know-how to finish a task. Each application has its own UI conventions and operation logic, and some tasks require specialized skills (e.g., spreadsheet formulas).

**Task:** In the "Features" slide, insert a table with 5 rows and 2 columns.

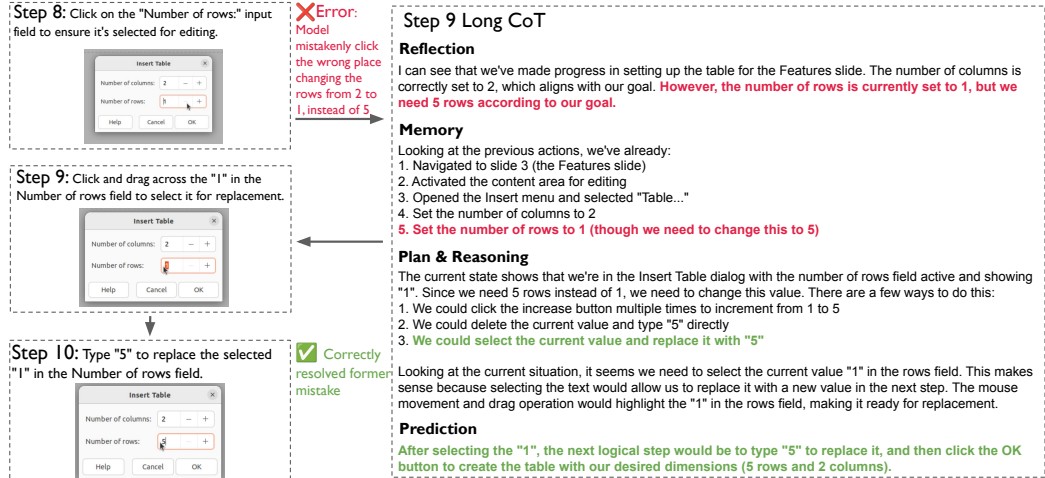

Figure 11: Reflective Long CoT Example: Before predicting the actual action, the model first reason according to the history and current action with reflection, memory, plan&reasoning and prediction in the CoT. The model identifies the former mistake and correct it in the later steps.

- Example 1: "I have a lookup table for the officers of each branch. Please fill the second table using VLOOKUP." The agent does not know the VLOOKUP function and therefore fails.

- Example 2: "Fill all the blank cells with the value in the cell above." The agent does not know the bulk-fill feature and instead edits cells one by one.

**2. High-precision grounding errors** Tasks that demand pixel-accurate actions frequently fail.

- Example: "Change the 2 in 'H2O' to a subscript." The agent must precisely drag-select only the "2," but often selects extra characters. Such fine-grained, letter-level grounding data are also hard to synthesize.

**3. Action repetition** When an incorrect action has no observable effect, the agent may keep predicting the same incorrect step. Occasionally it recovers, but often it loops indefinitely.

**4. Termination misjudgment** Sometimes the agent fails to notice that the task is already complete and continues acting, causing failure due to excessive extra actions. In other cases, it incorrectly assumes success and terminates prematurely.

**5. Long-horizon task failures** OSWorld includes tasks requiring >30–50 gold actions. Maintaining coherent context over so many steps remains challenging. For example: "Organize my desktop by placing academic papers in 'Paper_reading', coding projects in 'Projects', and everything else in 'Miscellaneous'. For files without clear names, determine the category by content."

**6. Insufficient error perception and recovery** Although the agent can detect some mistakes and reflect, the agent is still not good at perceive error as human does. For example, high-precision edits still pose problems. It may insert text one character off yet judge the action correct, and it often lacks a reliable strategy to undo and retry after mistakes.

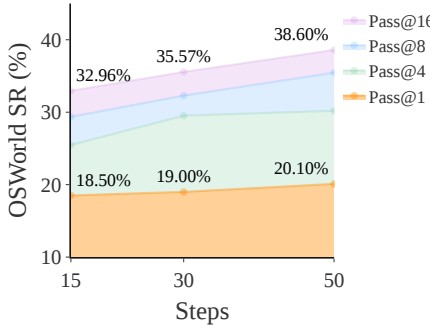

Figure 12: OSWorld Pass@N performance curves of OPENCUA-QWEN2-7B, temperature=0

### D.5 Extra analysis

**Agent model is not robust: small variance in the environment affects the task result.** As illustrated in Figure 12, OPENCUA-QWEN2-7B 's OSWorld performance (Pass@N) under temperature=0 exhibits significant outcome divergence despite nearly identical initial states—with only minor variations (e.g., system date). The curves for Pass@16 (38.60% SR at 50 steps) and Pass@1 (20.10% SR) demonstrate a >18% absolute gap, highlighting how minimal initial perturbations propagate into starkly different trajectories. This underscores the model's sensitivity to initial conditions even in deterministic (temp=0) settings, suggesting that seemingly trivial factors (e.g., temporal context) may critically influence multi-step reasoning.

# E OSWorld-Verified Results Detail

Table 7: Evaluation results of OPENCUA MODELS on **OSWorld-Verified**. OPENCUA-32B and OPENCUA-7B have been run for 3 turns. Pass@3 success rate is also calculated.

| Model | Setting | #Step | Succ. Rate | Succ. Len | Fail Len | Calc | Impr | Writer | Chrome | VLC | TB | OS | GIMP | VSCode | Multi | Total Succ. |
|---|---|---|---|---|---|---|---|---|---|---|---|---|---|---|---|---|
| OPENCUA-72B-PREVIEW | Turn 1 | 15 | 39.03% | 7.95 | 12.96 | 13 | 19 | 13 | 25 | 4 | 7 | 12 | 21 | 14 | 13 | 141 |
| | Turn 1 | 50 | 44.89% | 12.86 | 27.55 | 15 | 21 | 14 | 27 | 7 | 8 | 11 | 20 | 17 | 21 | 161 |
| | Turn 1 | 100 | 46.08% | 16.30 | 39.26 | 17 | 25 | 13 | 26 | 7 | 9 | 14 | 20 | 16 | 19 | 166 |
| | Turn 2 | 100 | 43.85% | 17.70 | 41.25 | 18 | 20 | 13 | 23 | 5 | 8 | 14 | 19 | 16 | 22 | 158 |
| | Turn 3 | 100 | 45.05% | 17.38 | 42.25 | 15 | 23 | 13 | 23 | 7 | 9 | 16 | 19 | 16 | 21 | 162 |
| | Avg. | 100 | 44.99% | 17.13 | 40.92 | 16.67 | 22.67 | 13.00 | 24.00 | 6.33 | 8.67 | 14.67 | 19.33 | 16.00 | 20.67 | 162.00 |
| | Pass@3 | 100 | **53.02%** | 19.01 | 39.02 | 20 | 29 | 15 | 27 | 8 | 9 | 19 | 20 | 17 | 28 | 192 |
| OPENCUA-32B | Turn 1 | 15 | 28.29% | 7.34 | 12.79 | 7 | 14 | 8 | 17 | 4 | 6 | 10 | 15 | 10 | 10 | 101 |
| | Turn 2 | 15 | 30.56% | 7.34 | 12.85 | 5 | 15 | 8 | 21 | 5 | 8 | 13 | 14 | 12 | 9 | 110 |
| | Turn 3 | 15 | 30.28% | 7.31 | 12.67 | 7 | 14 | 11 | 18 | 4 | 6 | 13 | 14 | 14 | 8 | 109 |
| | Avg. | 15 | 29.71% | 7.33 | 12.77 | 6.33 | 14.33 | 9.00 | 18.67 | 4.33 | 6.67 | 12.00 | 14.33 | 12.00 | 9.00 | 106.67 |
| | Pass@3 | 15 | **37.34%** | 7.83 | 12.85 | 9 | 19 | 11 | 25 | 6 | 8 | 13 | 18 | 14 | 12 | 135 |
| OPENCUA-32B | Turn 1 | 50 | 33.89% | 10.40 | 23.79 | 6 | 18 | 9 | 20 | 4 | 9 | 11 | 19 | 10 | 16 | 122 |
| | Turn 2 | 50 | 33.43% | 10.53 | 24.62 | 7 | 16 | 10 | 19 | 5 | 8 | 12 | 18 | 13 | 12 | 120 |
| | Turn 3 | 50 | 35.28% | 11.29 | 24.42 | 6 | 20 | 9 | 21 | 4 | 7 | 14 | 17 | 15 | 14 | 127 |
| | Avg. | 50 | 34.20% | 10.74 | 24.28 | 6.33 | 18.00 | 9.33 | 20.00 | 4.33 | 8.00 | 12.33 | 18.00 | 12.67 | 14.00 | 123.00 |
| | Pass@3 | 50 | **45.58%** | 122.33 | 24.34 | 11 | 22 | 12 | 29 | 6 | 9 | 15 | 23 | 17 | 21 | 165 |
| OPENCUA-32B | Turn 1 | 100 | 33.89% | 12.86 | 32.52 | 11 | 15 | 8 | 16 | 6 | 7 | 12 | 18 | 12 | 17 | 122 |
| | Turn 2 | 100 | 35.00% | 10.86 | 29.90 | 8 | 18 | 8 | 19 | 5 | 6 | 11 | 18 | 16 | 17 | 126 |
| | Turn 3 | 100 | 35.75% | 11.13 | 29.47 | 7 | 20 | 9 | 20 | 6 | 8 | 15 | 16 | 15 | 12 | 128 |
| | Avg. | 100 | 34.88% | 11.62 | 30.63 | 8.67 | 17.67 | 8.33 | 18.33 | 5.67 | 7.00 | 12.67 | 17.33 | 14.33 | 15.33 | 125.33 |
| | Pass@3 | 100 | **45.10%** | 13.02 | 33.27 | 12 | 23 | 10 | 27 | 7 | 8 | 15 | 20 | 19 | 22 | 163 |
| OPENCUA-7B | Turn 1 | 15 | 26.18% | 7.70 | 12.73 | 5 | 17 | 6 | 17 | 5 | 8 | 7 | 13 | 10 | 6 | 94 |
| | Turn 2 | 15 | 23.89% | 6.97 | 12.77 | 4 | 14 | 7 | 13 | 3 | 6 | 10 | 12 | 10 | 7 | 86 |
| | Turn 3 | 15 | 23.06% | 6.94 | 12.92 | 4 | 12 | 5 | 15 | 4 | 6 | 8 | 10 | 10 | 9 | 83 |
| | Avg. | 15 | 24.38% | 7.20 | 12.81 | 4.33 | 14.33 | 6.00 | 15.00 | 4.00 | 6.67 | 8.33 | 11.67 | 10.00 | 7.33 | 87.67 |
| | Pass@3 | 15 | **30.74%** | 7.77 | 12.82 | 6 | 19 | 7 | 20 | 5 | 8 | 11 | 13 | 11 | 11 | 111 |
| OPENCUA-7B | Turn 1 | 50 | 28.81% | 10.06 | 24.14 | 7 | 13 | 8 | 18 | 4 | 7 | 11 | 13 | 10 | 13 | 104 |
| | Turn 2 | 50 | 27.73% | 8.93 | 24.77 | 5 | 17 | 8 | 18 | 5 | 6 | 10 | 10 | 11 | 9 | 99 |
| | Turn 3 | 50 | 27.93% | 8.12 | 24.34 | 7 | 16 | 7 | 17 | 5 | 6 | 9 | 11 | 11 | 11 | 100 |
| | Avg. | 50 | 28.16% | 9.04 | 24.42 | 6.33 | 15.33 | 7.67 | 17.67 | 4.67 | 6.33 | 10.00 | 11.33 | 10.67 | 11.00 | 101.00 |
| | Pass@3 | 50 | **35.75%** | 9.91 | 24.69 | 9 | 20 | 8 | 22 | 5 | 7 | 13 | 15 | 12 | 18 | 129 |
| OPENCUA-7B | Turn 1 | 100 | 27.30% | 9.79 | 28.71 | 4 | 16 | 6 | 18 | 5 | 6 | 10 | 14 | 10 | 9 | 98 |
| | Turn 2 | 100 | 26.04% | 9.68 | 27.55 | 5 | 13 | 6 | 18 | 5 | 6 | 10 | 11 | 10 | 10 | 94 |
| | Turn 3 | 100 | 26.67% | 9.61 | 27.20 | 6 | 16 | 7 | 14 | 5 | 6 | 10 | 12 | 11 | 9 | 96 |
| | Avg. | 100 | 26.67% | 9.69 | 27.82 | 5.00 | 15.00 | 6.33 | 16.67 | 5.00 | 6.00 | 10.00 | 12.33 | 10.33 | 9.33 | 96.00 |
| | Pass@3 | 100 | **36.48%** | 11.73 | 30.07 | 7 | 22 | 10 | 23 | 5 | 7 | 13 | 17 | 12 | 16 | 132 |
| OPENCUA-A3B | Turn 1 | 15 | 16.90% | 7.66 | 12.36 | 1 | 9 | 5 | 12 | 4 | 3 | 4 | 10 | 8 | 5 | 61 |
| | Turn 1 | 50 | 19.94% | 9.10 | 21.13 | 1 | 11 | 8 | 10 | 2 | 1 | 9 | 14 | 10 | 6 | 72 |
| | Turn 1 | 100 | 17.73% | 10.77 | 26.60 | 3 | 12 | 4 | 12 | 2 | 3 | 3 | 14 | 9 | 2 | 64 |
| OPENCUA-QWEN2-7B | Turn 1 | 15 | 19.94% | 7.25 | 12.74 | 4 | 11 | 8 | 9 | 5 | 6 | 4 | 10 | 12 | 3 | 72 |
| | Turn 1 | 50 | 20.61% | 8.54 | 21.28 | 3 | 12 | 10 | 15 | 5 | 7 | 2 | 6 | 10 | 4 | 74 |
| | Turn 1 | 100 | 23.06% | 9.73 | 26.19 | 5 | 14 | 10 | 12 | 4 | 5 | 6 | 10 | 11 | 6 | 83 |

# F Dataset Statistics, Annotation Details, and AgentNetTool Details

## F.1 AGENTNET TOOL

Efficient and accurate annotation is essential for collecting high-quality computer-use agent data, yet no existing tools support natural, cross-platform task recording by non-technical users. To address this, we developed a user-friendly annotation tool that streamlines the collection and verification of computer-use demonstrations (Figure 13), runs on annotators' personal computers and records demonstrations in the background, capturing: (1) screen videos, (2) mouse and keyboard signals, and (3) accessibility trees (Axtree). These data are

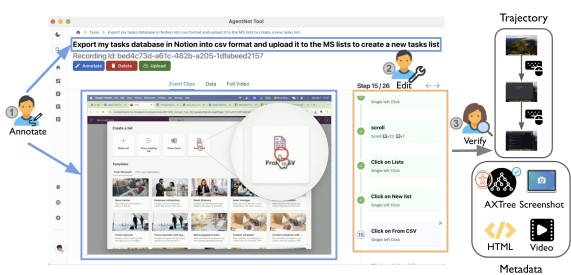

Figure 13: AGENTNET TOOL annotation and verification.

then processed into state-action trajectories (see details below in Section F.1), allowing annotators to review, edit, and submit demonstrations along with task instructions describing the overall goal. Former works require the annotators to demonstrate "gold" trajectories with all-correct steps, but this actually limits model's capability to detect and recover from errors. We believe that annotation error is not all bad, as long as we can identify and utilize them (see Section 3.1), so we relax the requirement of all correct actions. Our implementation leverages several established tools: mouse and keyboard input tracking is based on DuckTrack [36] and OpenAdapt [27]; screen recording utilizes OBS Studio [26]; and accessibility tree (Axtree) parsing follows the OSWorld framework [44]. Additional implementation details can be found in Appendix F.1.1.

**Annotation pipeline** We designed our data collection with two key goals: **diversity** and **complexity**. Annotators were provided a curated list of around 200 applications and websites spanning various domains and were encouraged to demonstrate complex workflows involving professional features or multi-app interactions. Tasks were required to have more than 15 steps; those with <5 steps were rejected. To ensure wide coverage and real-world authenticity, we recruited annotators from both crowd-sourcing platforms and annotation companies. All annotators signed consent forms, and we use a multi-layer privacy protection mechanism to safeguard user data (Appendix F.1.2). To study model generalization, we split data into Windows/macOS and Ubuntu, ensuring no overlap with OSWorld tasks to prevent data leakage. All tasks were manually verified and labeled as *rejected*, *ok*, *good*, or *excellent* based on goal clarity, diversity, and complexity. Other annotation details are provided in Appendix F.3.

**Constructing compact state-action trajectories** Raw demonstrations consist of high-frequency screen recordings and fine-grained interaction signals (mouse movements, clicks, scrolls, key presses). A typical task can produce thousands of low-level actions that are too dense and inefficient for training. To address this challenge, we developed techniques including action reduction and state-action matching to construct compact state-action pairs $\langle s_i, a_i \rangle$. **(1) Action reduction**: We developed a rule-based method to compress and reduce these dense action signals into a smaller set of meaningful actions while preserving essential action information. We first compress atomic signals into higher-level operations. Mouse move events are treated as preconditions for clicks or drags, and only their start and end positions are retained. Scrolls are merged into single-directional actions with accumulated wheel counts. Consecutive key presses are merged into text input strings, while modifier combinations (e.g., CTRL+C) are abstracted into hotkey actions. We also combine common multistep gestures such as drags or double-clicks. This process yields a streamlined action sequence aligned with the pyautogui action space, as shown in Table 9. **(2) State-action matching**: To pair each action $a_i$ with a representative state $s_i$, we extract keyframes from the screen recording that capture the system state *immediately before* the action occurs. However, naively aligning keyframes to action timestamps of mouse clicks risks leaking future information; e.g., the mouse may already be positioned over a button, making the prediction trivial. To address this challenge, for mouse clicks, we backtrack to the beginning of the mouse's pre-movement phase and search backward to find the last visually distinct frame.After the final action, we append a terminal frame along with a corresponding termination action.

### F.1.1 Tool Features

1. **Action Reduction**: We use tools like `pynput` to capture users' atomic actions. These atomic actions are then reduced to semantically meaningful actions, such as 'click', 'key_press', 'key_release', 'type', 'drag', 'move', and 'scroll'. This reduction enables models to more effectively learn from human demonstrations and allows annotators and verifiers to understand trajectories more easily.

2. **A11y Tree and HTML Processing**: To obtain textual representations of observations, we implement efficient fetching and processing mechanisms for accessibility (a11y) trees and HTML. For the a11y tree, we apply pruning rules to select only essential element attributes, ensuring the fetching process for each tree takes minimal time. For HTML, we develop a browser plugin that automatically captures the HTML structure of websites.

3. **Element Localization**: To help users verify the correctness of their actions, we extract text associated with click locations. Using the a11y tree or HTML, we fetch the bounding box most likely clicked and extract textual information from it. If the extracted text is insufficient, we leverage GPT to predict the semantic information of the clicked element.

4. **Trajectory Visualization**: We design a user-friendly interface to ensure a seamless annotation experience. For each action, we display its description, a corresponding video clip, and the a11y tree. Additionally, we provide the full video of the entire trajectory for better context.

5. **Verification and Administration Systems**: To ensure the quality of the collected data, we develop verification and administration systems that streamline the process of validating annotations and maintaining dataset integrity.

### F.1.2 Privacy Protection

We implemented a multi-layer privacy protection framework in our data collection process. First, annotators must agree to a consent form that clearly states the scope of data collection, including screen recordings, actions, and system information. The form explicitly prohibits recording private or sensitive information. The tool is designed with privacy-first principles: no data is transmitted to servers without manual upload by annotators, and annotators can review all collected data (including videos, actions, and accessibility tree structures) before submission. We further ensure privacy through a two-stage verification process: manual review by internal team members during task verification, and automated examination of the task trajectory using `GPT-4o` during post-processing. Tasks containing private information are rejected immediately.

**1. GPT-Based Privacy Analysis**

**Data Ingestion:**     The system loads task descriptions and step-by-step user actions (*Observations*, *Thoughts*, *Action Descriptions*, etc.) from JSON. These records provide details of users' intent, the interface elements users interacted with, and any textual or visual cues relevant to the task.

**GPT Inference:**     The script calls OpenAI's API with a carefully structured prompt, requesting GPT to produce a privacy classification in one of four levels: *None*, *Low*, *Medium*, or *High*. By passing the user's detailed action steps and observations to GPT, the system gathers a structured output that includes an explicit `privacy_sensitivity` label.

GPT accelerates the classification process, effectively scanning large volumes of user trajectories with consistent logic. It can identify and label potentially sensitive fields or behaviors with minimal human oversight. However, GPT's inherent reliance on keywords sometimes causes it to mark a scenario as *High*, even if the user simply viewed (but did not fill) a password field. Such errors underscore the necessity of post-processing or additional contextual checks to distinguish between *potential* versus *actual* private data entry.

**2. Human Verification**     Human reviewers examine the same recordings or textual records to determine whether genuinely confidential information was entered. For example: If only an email address is mentioned, human reviewers might label it as *Low* or *None*. If a direct password or banking details appear, human reviewers assign *High*.

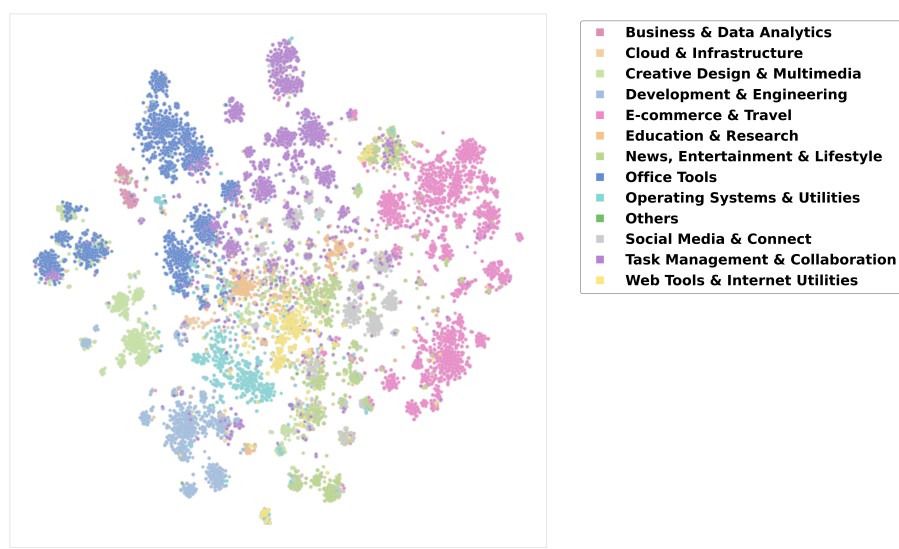

Figure 14: Clustering and t-SNE visualization of all task domains.

In "false alarm" cases (for example, a password field is displayed but left blank), humans typically label them as *None*. This nuanced reasoning often yields higher precision but comes at the cost of increased labor and time. Furthermore, variability in human judgments may occur if guidelines are not strictly enforced.

## F.2 AGENTNET Statistics and Analysis

### F.2.1 Diversity

**Task Domains** We categorize the collected tasks into 4 main domains and 11 subdomains based on their topics, involved applications and actions in the tasks (Figure 3). Table 8 lists representative applications for each domain. To label each task trajectory, we leveraged GPT-4o to complete the classification by representing each task using the task instruction and L1-level CoT. We manually examined 200 tasks randomly and the classification accuracy is over 96%. We then embedded the task trajectories using OpenAI's text-embedding-3-small model and visualize them t-SNE visualization in Figure 14. Interestingly, the layout mirrors typical computer-usage patterns: for instance, Office Tools cluster near Business & Data-Analytics, while E-commerce & Travel sit close to Social-Media & Connect on the opposite side of the map. Finally, we chose 100 representative tasks around the cluster centroids to form our offline benchmark, AGENTNETBENCH.

| Domain | App/Web |
|---|---|
| E-commerce & Travel | Amazon.com, Booking.com |
| News, Entertainment & Lifestyle | Spotify, Netflix |
| Social Media & Communication | WhatsApp, Instagram |
| Office Tools | Microsoft Office, Google Docs |
| Task Management & Collaboration | Zoom, Gmail, Slack |
| Creative Design & Multimedia | Photoshop |
| Development & Engineering | VSCode, PyCharm, Git |
| Knowledge Discovery & Research | Google Scholar, ResearchGate |
| Data Analysis, Business & Cloud | Tableau, Power BI, AWS |
| Web Tools & Internet Utilities | Chrome Extensions |
| Operating Systems & Utilities | Finder, Activity Monitor |

Table 8: Example App/Web by Doman

**Applications and Websites**   Applications and websites are tracked using the AgentNet Tool. Specifically, application names are captured by recording the process name when a user opens an application, while website URLs are recorded through our browser plugin. Discrepancies in process names across different operating systems and different versions are resolved using GPT, achieving an accuracy of up to 83% with this combined method. Based on these results, we analyzed the distribution of the dataset across applications and websites. Web browsers account for a significant portion of the dataset, as nearly half of the data involves web applications. The results indicate that the dataset encompasses over 140 applications and 190 mainstream websites.

### F.2.2   Complexity

Our collected tasks average 18.6 steps per task. We evaluate task complexity across five dimensions: multi-application/website usage, professional knowledge requirement, uncommon feature usage, repetitive simple subtasks, and logically coherent long sequences. Based on GPT-4o analysis, 30.6% tasks require multiple applications/websites, 12.9% involve professional knowledge, and 12.9% use uncommon features in Figure 16. Then we asked GPT to rate the complexity of tasks on a 1-10 scale, where 1 represents basic operations like file opening, and 10 indicates complex tasks requiring multiple steps, domain knowledge, or sophisticated reasoning. The complexity distribution is shown in Figure 15. It can be seen that most of the tasks have a medium or high level of complexity.

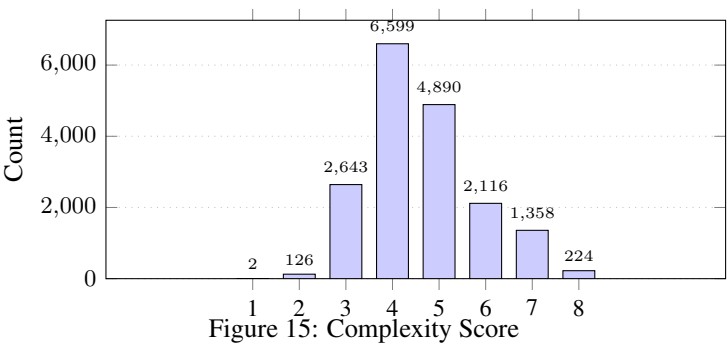
Figure 15: Complexity Score

### F.2.3   Action distribution

| Human Action | Action Description | Agent Action |
|---|---|---|
| Click | Click at a specific position | **click**(x, y, button) |
| Middle Click | Middle click at a specific position | **middleClick**(x, y) |
| Double Click | Double click at a specific position | **doubleClick**(x, y, button) |
| Triple Click | Triple click at a specific position | **tripleClick**(x, y, button) |
| Mouse Move | Move mouse to a specific position | **moveTo**(x, y) |
| Drag | Drag mouse from one position to another | **dragTo**(x, y) |
| Scroll | Scroll vertically or horizontally | **scroll**(dx, dy) / **hscroll**(dx, dy) |
| Type | Type a string of text | **write**(text) |
| Press | Press a specific key | **press**(key) |
| Hotkey | Perform a combination of keys | **hotkey**(key1, key2) |
| Wait | Wait for a few seconds | **wait**() |
| Terminate | End the task with success or failure | **terminate**('success' or 'failure') |

Table 9: Overview of Human Actions and Corresponding Agent Action Functions

Table 10 summarizes the action frequencies across 3 systems. We counted the action distribution of 41,428 computer-use tasks, including Ubuntu 23,797 tasks, Windows 12,431, and macOS 5,200. Click is the dominant action—more than 60% on three systems. Hardware and usage patterns drive the secondary behaviors: macOS trackpads lead to heavier vertical/horizontal scrolling and more hotkey use; Windows mouse workflows show higher proportions of right-click and middle-click; and Ubuntu's keyboard-centric, terminal-oriented culture results in the greatest shares of text input.

Table 10: Action–type distribution (%) per operating system.

| System | click | doubleClick | dragTo | hotkey | hscroll | middleClick | moveTo |
|---|---|---|---|---|---|---|---|
| Windows | 60.03 | 1.99 | 4.35 | 1.52 | 0.00 | 0.01 | 8.95 |
| macOS | 57.22 | 2.20 | 3.48 | 2.48 | 0.46 | 0.00 | 9.02 |
| Ubuntu | 63.62 | 3.19 | 2.30 | 2.21 | 0.00 | 0.00 | 4.53 |

| System | press | rightClick | scroll | terminate | tripleClick | write |
|---|---|---|---|---|---|---|
| Windows | 5.27 | 1.10 | 4.51 | 4.42 | 0.00 | 7.85 |
| macOS | 6.39 | 1.01 | 5.52 | 4.40 | 0.08 | 7.73 |
| Ubuntu | 3.78 | 2.40 | 2.06 | 5.89 | 0.22 | 9.81 |

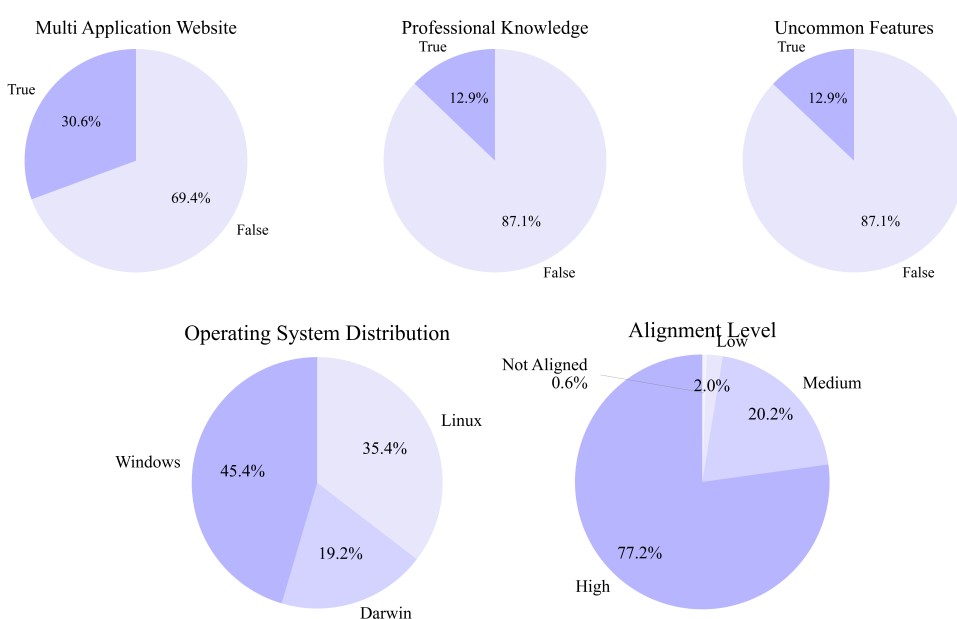

Figure 16: Distributions of data characteristics: presence of multi-application websites, inclusion of professional knowledge, presence of uncommon features, source operating systems and alignment levels.

### F.3 Annotation Details: Annotation Strategy, Annotator Source and Cost

**Annotation Strategy** We first surveyed the most popular websites and applications across a wide range of domains—entertainment, office tools, and more. We selected the 200 + most widely used ones. Tasks were not pre-assigned, because annotators' familiarity varies (especially with professional tools). We set a limit number for the apps and allowed annotators to choose. Annotators could also brainstorm new tasks with provided relevant YouTube tutorials so they could explore and create additional tasks.

**Annotator Source** We recruited annotators from four sources: internal students, external university students, annotation companies, and crowd-sourcing platform - Prolific. Table 11 shows the distribution of annotators and tasks. While annotators from Prolific and Company1 were native English speakers, others were native Chinese speakers. All annotators were required to document task goals in English and try to use English system settings, applications and websites to ensure broader applicability. (The numbers in the table are annotated tasks before verification)

**Annotation and CoT synthesis cost** Annotating 22K tasks takes 6 months. All annotators are part-time. The total annotation cost was about USD 20,000. Annotation speed is roughly ten tasks

| Annotator Source | Accepted Uploads | Annotator Count |
|---|---|---|
| Internal Students | 4943 | 38 |
| External Students | 5168 | 135 |
| Prolific | 1218 | 294 |
| Company1 | 2235 | 72 |
| Company2 | 3556 | 51 |
| Company3 | 1975 | 14 |
| Company4 | 5000 | 30 |
| **Total** | **24095** | **634** |

Table 11: Annotation Source Statistics (before verification)

per hour. The cost of synthesizing CoT costs USD 0.6 per task on average. The total cost of building this dataset is about USD 32,000.

# G   OSWorld Case Example

The trajectory outlines the process of installing a manually developed Chrome extension located in the Desktop directory into the Google Chrome browser for testing or usage purposes. Notably, the step labeled in red highlights the agent's capacity for reflection and error correction—it initially diverges into an incorrect path, subsequently recognizes the mistake, and successfully reorients itself to proceed with the correct course of action.

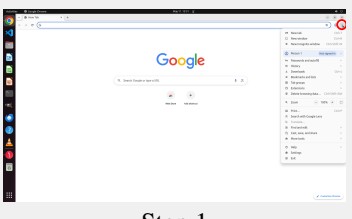

**Step 1**

**Thought:** I need to install a Chrome extension that's located in the Desktop directory. Looking at the current state, I'm on the Google homepage in Chrome. To install an extension, I need to access the Chrome Web Store or use the extensions menu.

I can see that there's a "Web Store" shortcut on the Google homepage which could be useful. However, I also notice the Chrome menu button (three vertical dots) in the top-right corner of the browser. This menu would give me access to more Chrome settings and features, including extensions.

Since I need to install a local extension file, the most direct approach would be to: 1. Click on the Chrome menu (three dots) 2. Navigate to "More tools" > "Extensions" or directly to "Chrome Web Store" 3. Upload the extension file from the Desktop.

The Chrome menu button in the top-right corner is the logical next action because it provides access to all Chrome settings and features, including extensions management. Clicking this button will open a dropdown menu with various options, including those related to extensions.

I anticipate that after clicking the menu button, I'll see options that will allow me to navigate to the extensions page or Chrome Web Store, where I can then upload the extension file from the Desktop.

**Action:** Click on the three-dot menu button in the top-right corner of the Chrome browser window to open the Chrome menu.

**Code**: pyautogui.click(x=0.988, y=0.081)

- - - - - - - - - - - - - - - - - - - - - - - - - - - - - - - - - - - - - - - - - - - - - - - -

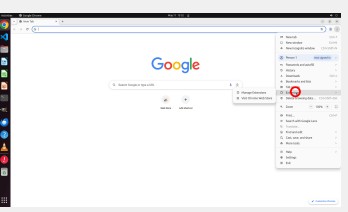

**Step 2**

**Thought:** After clicking on the three-dot menu in the previous step, the Chrome menu has successfully opened, displaying various options. This is exactly what I expected to happen, so we're making good progress toward our goal of installing a Chrome extension from the Desktop. Looking at the current state, I can see that the Chrome menu is now open with all its options. To install a Chrome extension, I need to access the Extensions page or use the Web Store. Based on the goal of installing an extension from the Desktop directory, I should look for the Extensions option in this menu.

I can see that there is an "Extensions" option in the menu, which is exactly what I need. Clicking on this option will take me to Chrome's Extensions page, where I can then load an unpacked extension from the Desktop directory.

The logical next action would be to click on the "Extensions" option in the Chrome menu. This will navigate to the Extensions management page where I can then find an option to load an unpacked extension. From there, I'll be able to browse to the Desktop directory and select the extension file for installation.

By clicking on the Extensions option, I expect to be taken to the Extensions management page where I'll see a list of currently installed extensions and options to manage them, including an option to "Load unpacked" which I'll need to use to install the extension from the Desktop.

**Action:** Click on the "Extensions" option in the Chrome menu to access the extensions management page.

**Code:** pyautogui.click(x=0.867, y=0.406)

- - - - - - - - - - - - - - - - - - - - - - - - - - - - - - - - - - - - - - - - - - - - - - - -

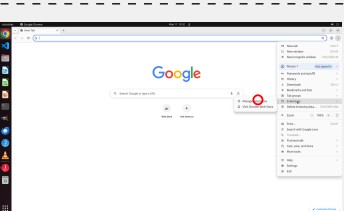

**Step 3**

**Thought:** After clicking on the "Extensions" option in the Chrome menu, I can see that the menu has expanded to show more options related to extensions. This is the expected result of the previous action.

Looking at the current state, I can see that there are two main options related to extensions: 1. "Manage Extensions" - This would take me to the page where I can manage currently installed extensions 2. "Visit Chrome Web Store" - This would take me to the Chrome Web Store where I can download extensions from the official store

Since our goal is to install a Chrome extension that is located on the Desktop directory, I need to access the extensions management page first. From there, I would be able to access the "Load unpacked" button which allows installing extensions from local files rather than from the Chrome Web Store.

The logical next action would be to click on "Manage Extensions" since this is the option that will take me to the extensions management page where I can install the extension from the Desktop. This is the most direct path to achieving our goal of installing a locally developed extension.

By clicking on "Manage Extensions," I expect to be taken to a page that shows all currently installed extensions and provides options to add new extensions, including the "Load unpacked" option that I'll need to use to install the extension from the Desktop directory.

**Action:** Click on the "Manage Extensions" option in the Chrome menu to access the extensions management page.

**Code:** pyautogui.click(x=0.755, y=0.403)

- - - - - - - - - - - - - - - - - - - - - - - - - - - - - - - - - - - - - - - - - - - - - - - -

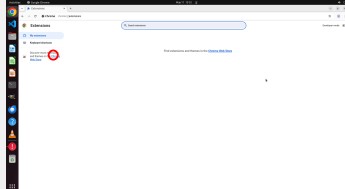

**Step 4**

**Thought:** After clicking on "Manage Extensions" in the previous step, I can see that I've successfully navigated to the Chrome extensions management page. This is exactly where I need to be to continue with the installation of the new Chrome extension.

Looking at the current state, I can see that I'm on the chrome://extensions page which shows the list of currently installed extensions. However, I don't see any extensions listed yet, which suggests either there are no extensions installed or the page needs to be refreshed to show them.

To install a new extension manually from a local file on the Desktop, I need to: 1. Access the Chrome Web Store or a similar interface where I can upload or load an extension from a file 2. Navigate to the extension file location (Desktop) 3. Install the extension

I notice there are a few options available to me: - I could click on "My extensions" to see if there are any installed extensions - I could click on "Keyboard shortcuts" to configure shortcuts - I could click on the "Chrome Web Store" link to browse the store - I could click on the "Chrome" link in "Discover more extensions and themes on the Chrome Web Store"

Since my goal is to install a locally developed extension, I need to find an option that would allow me to load an extension from a file. The "Chrome Web Store" link seems to be the most logical next step, as it would likely provide access to extension management functions including loading local extensions.

Clicking on the "Chrome" link in the text "Discover more extensions and themes on the Chrome Web Store" would take me to the Chrome Web Store, where I might find options to access developer tools or load extensions from local files. This seems like the most logical next action to progress toward my goal of installing the extension from the Desktop directory.

**Action:** Click on the "Chrome" link in the text "Discover more extensions and themes on the Chrome Web Store"

**Code:** pyautogui.click(x=0.138, y=0.284)

---

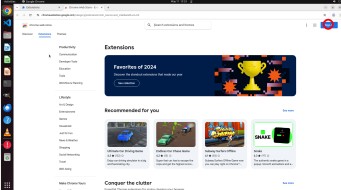

**Step 5**

**Thought:** After clicking on the "Chrome" link in the Extensions management page, I've been taken to the Chrome Web Store. This is a good start, but I notice that I'm currently in the Extensions category of the Chrome Web Store, which is for browsing and downloading extensions from the store, not for installing local extensions.

To install a Chrome extension from a local file (located in the Desktop directory as mentioned in the goal), I need to access the Chrome Web Store's developer page or use a different approach. Looking at the current state, I need to navigate to a different section that allows for installing unpacked extensions.

I can see that there's a "Sign In" button in the top-right corner of the page. Signing in to Chrome is often necessary for extension development and installation. Additionally, there might be a way to access developer options or a special section for installing unpacked extensions after signing in. Given the goal of installing a locally developed Chrome extension from the Desktop directory, the next logical step would be to sign in to Chrome. This would likely give me access to more advanced options for extension installation, including the ability to load an unpacked extension from a local directory.

After signing in, I expect to either find a direct option to load an unpacked extension or need to upload the extension file from the Desktop. Signing in is a necessary step in the process of manually installing a Chrome extension from a local directory.

**Action:** Click on the "Sign In" button in the top-right corner of the Chrome Web Store page.

**Code:** pyautogui.click(x=0.954, y=0.131)

---

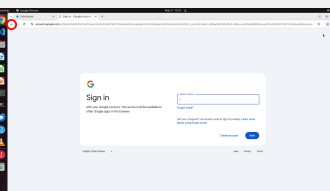

**Step 6**

**Thought:** After clicking the "Sign In" button in the Chrome Web Store, I've been redirected to a Google sign-in page. This makes sense as many Chrome extensions require a signed-in profile to be installed properly. The sign-in page is asking for Google account credentials to proceed.

Looking at my progress toward the goal of installing a Chrome extension manually from the Desktop directory, I need to: 1. Access the Chrome Web Store (completed) 2. Sign in to Chrome (in progress) 3. Navigate to the extension file on the Desktop 4. Install the extension manually

I'm currently at step 2, where I need to sign in to a Google account to proceed. However, I notice that I might not need to complete the entire sign-in process. Since the goal is to install a locally downloaded extension, I might be able to go back and find an option to load an extension from a local file without fully signing in.

Looking at the browser interface, I see the back button in the top-left corner of the browser. Clicking this back button would allow me to return to the Chrome Web Store page where I might find alternative options for installing an extension from a local file, such as a "Load unpacked" button that's typically available when you're in the extensions management page.

The logical next action would be to click the back button to return to the Chrome Web Store. This would allow me to potentially find a more direct way to install the extension from the Desktop without completing the sign-in process. After clicking back, I expect to return to the Chrome Web Store page where I can look for options related to loading local extensions.

**Action:** Click the back button in the top-left corner of the Chrome browser window to return to the previous page.

**Code:** pyautogui.click(x=0.048, y=0.086)

---

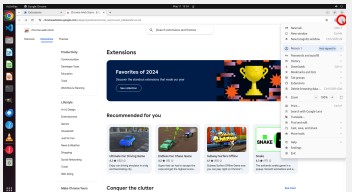

**Step 7**

**Thought:** After clicking the back button, I've successfully returned to the Chrome Web Store page from the Google sign-in page. This was the right move since signing in wasn't necessary to install a local extension.

Now I need to find a way to install the extension from the Desktop directory. Looking at the current state of the Chrome Web Store page, I need to find an option that would allow me to load an extension from a local file. This functionality is typically available through a "More" menu or similar option on the Chrome Web Store page.

I can see that there's a three-dot menu button in the top-right corner of the Chrome browser window. This is likely where I'll find additional options for managing extensions, including the ability to load an extension from a local file. Clicking this button should reveal a dropdown menu with various options, potentially including "More Extensions" or "Load unpacked extension" which would allow me to navigate to the Desktop directory and select the extension file.

The logical next action is to click on this three-dot menu button in the top-right corner of the browser. This should open a menu with additional options related to extensions, which will help me progress toward manually installing the extension from the Desktop directory.

**Action:** Click on the three-dot menu button in the top-right corner of the Chrome browser window to access additional browser options.

**Code:** pyautogui.click(x=0.988, y=0.083)

---

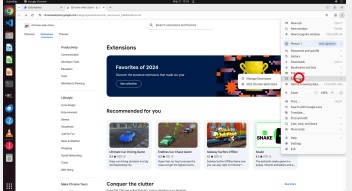

**Step 8**

**Thought:** After clicking the three-dot menu in the top-right corner of the Chrome browser, a dropdown menu has appeared with various browser options. This is exactly what I was expecting to see, as this menu provides access to Chrome's core functionality including extensions management. Looking at my progress toward the goal of installing a Chrome extension manually from the Desktop directory, I've successfully: 1. Opened the Chrome menu 2. Accessed the Extensions option 3. Navigated to the Chrome Web Store 4. Attempted to sign in (but determined it wasn't necessary) 5. Opened the Chrome menu again to access more options

Now I need to find the best way to install the extension from the Desktop. Since I already know the extension file is on the Desktop, I should look for an option that would allow me to load an unpacked extension or access the Desktop files.

Looking at the dropdown menu, I can see an "Extensions" option with a submenu indicator (>). This is exactly what I need to click next because: 1. It will take me to the Extensions management page or provide extension-related options 2. From there, I should be able to find an option to load an unpacked extension 3. This is the standard way in Chrome to manage extensions, including loading them from local files

Clicking on the "Extensions" menu item will likely show me additional options such as "Load unpacked" which would allow me to navigate to the Desktop and select the extension folder. This is the most direct path to manually installing the extension from the Desktop as required.

**Action:** Click on the "Extensions" option in the Chrome menu to access extension management functions.

**Code:** pyautogui.click(x=0.863, y=0.406)

- - - - - - - - - - - - - - - - - - - - - - - - - - - - - - - - - - - - - - - - - -

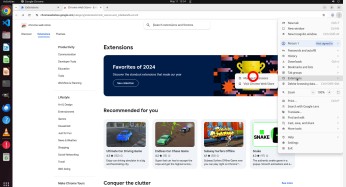

**Step 9**

**Thought:** After clicking on the "Extensions" option in the Chrome menu, a submenu has appeared with two options: "Manage Extensions" and "Visit Chrome Web Store". This is exactly what I need to continue with the installation process.

Looking at my progress toward installing the Chrome extension: 1. I opened the Chrome menu and accessed Extensions 2. I went to the Chrome Web Store's Extensions page 3. I attempted to sign in but decided to go back 4. I opened the Chrome menu again and clicked on Extensions 5. Now I'm seeing a submenu with extension-related options

To install a local extension from the Desktop directory, I need to access the extension management interface. The "Manage Extensions" option is exactly what I need, as it will allow me to access the extension installation functionality. This is where I'll be able to load an unpacked extension from a local directory.The "Manage Extensions" option is the first item in the submenu that has appeared. Clicking on this option will take me to the extensions management page where I can find the option to add an unpacked extension. From there, I'll be able to navigate to the Desktop directory and select the extension file to complete the installation.The logical next action is to click on "Manage Extensions" in the submenu. This will take me to the extensions management page where I can proceed with installing the extension from the Desktop directory.

**Action:** Click on the "Manage Extensions" option in the Chrome menu submenu.

**Code:** pyautogui.click(x=0.738, y=0.401)

- - - - - - - - - - - - - - - - - - - - - - - - - - - - - - - - - - - - - - - - - -

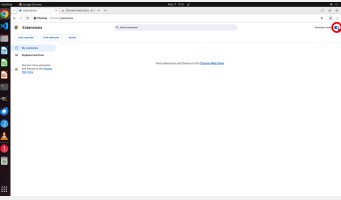

**Step 10**

**Thought:** After clicking on "Manage Extensions" in the Chrome menu, I've been taken to the Chrome extensions management page. This is exactly where I need to be to continue with the installation of the new Chrome extension.

Looking at my progress toward the goal of installing a Chrome extension manually from the Desktop directory: 1. I've successfully accessed the Chrome browser 2. I've navigated to the extensions management page 3. I now need to find a way to load the extension from the Desktop The extensions page shows several options, including "My extensions" and "Keyboard shortcuts," but I don't see a direct option to upload or add an extension from a file. However, I notice the "Developer mode" toggle in the top-right corner of the page. This is exactly what I need, as Developer mode typically provides additional options for developers, including the ability to load unpacked extensions.To install a manual extension, I need to enable Developer mode first. Once Developer mode is enabled, I should get additional options that will allow me to load an unpacked extension from a local directory like the Desktop.The logical next action is to click on the "Developer mode" toggle switch in the top-right corner of the page. This will enable developer options and should reveal additional controls that will allow me to load the extension from the Desktop directory. After clicking this toggle, I expect to see new buttons appear that will let me access the file system and select the extension file.

**Action:** Click on the "Developer mode" toggle switch in the top-right corner of the page to enable developer options.

**Code:** pyautogui.click(x=0.988, y=0.131)

- - - - - - - - - - - - - - - - - - - - - - - - - - - - - - - - - - - - - - - - - -

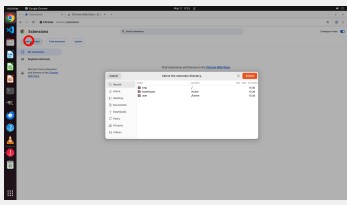

**Step 11**

**Thought:** After clicking on the "Developer mode" toggle switch, it has successfully been enabled as shown by the blue toggle in the top-right corner of the page. This has revealed additional functionality that wasn't visible before, including the "Load unpacked" button which is exactly what we need to install our extension manually.Now that Developer mode is enabled, I can see the "Load unpacked" button which allows us to install extensions from local files rather than from the Chrome Web Store. This is perfect for our goal since we need to install an extension that's located in the Desktop directory.Looking at the available options, I can see: 1. "Load unpacked" button - This is what we need to install our local extension 2. "Pack extension" button - Used for creating a packaged version of an extension 3. "Update" button - For updating existing extensions 4. The sidebar navigation showing "My extensions" and "Keyboard shortcuts"To install the extension from the Desktop directory, the logical next action is to click on the "Load unpacked" button. This will open a file browser dialog that will allow us to navigate to the Desktop and select the extension folder. Once we select the folder, Chrome will install the extension from the unpacked files.Clicking the "Load unpacked" button is the most appropriate action because: 1. It directly addresses our goal of installing a locally developed extension 2. It's the standard way to install extensions from local files in Chrome 3. The Developer mode is already enabled, which is a prerequisite for using this feature

After clicking this button, I expect a file browser dialog to appear, allowing us to navigate to the Desktop and select the extension folder.

**Action:** Click on the "Load unpacked" button to open a file browser dialog for selecting the extension folder from the Desktop.

**Code:** pyautogui.click(x=0.072, y=0.179)

- - - - - - - - - - - - - - - - - - - - - - - - - - - - - - - - - - - - - - - - - -

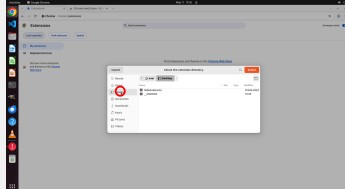

**Step 12**

**Thought:** After clicking the "Load unpacked" button, a file browser dialog has opened as expected. This dialog allows me to navigate to the location where the extension is stored, which according to our goal should be in the Desktop directory.

Looking at the current state, I can see that the file browser dialog is showing the contents of what appears to be the home directory, with folders like tmp, Downloads, and user visible. However, our goal is to install a Chrome extension that's located in the Desktop directory. Therefore, I need to navigate to the Desktop folder.

I notice that the Desktop folder is conveniently listed in the left navigation panel of the file browser. Since we need to access the Desktop to find the extension files, clicking on the Desktop folder in the left navigation panel is the logical next step. This will change the view to show the contents of the Desktop directory, where we should be able to find the extension files.

Once I click on Desktop, I expect the main panel to update and show all files and folders located on the Desktop. From there, I'll need to locate the specific extension folder and select it to complete the installation process.

The logical next action is to click on the "Desktop" option in the left navigation panel of the file browser dialog to navigate to the Desktop directory where our extension is located. **Action:** Click on the "Desktop" option in the left navigation panel of the file browser dialog.

**Code:** pyautogui.click(x=0.333, y=0.470)

- - - - - - - - - - - - - - - - - - - - - - - - - - - - - - - - - -

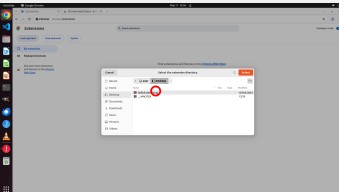

**Step 13**

**Thought:** After clicking on the "Desktop" option in the left navigation panel, the file browser dialog has now displayed the contents of the Desktop directory. This is exactly what we needed to proceed with installing the Chrome extension.

Looking at the current state, I can see that the Desktop directory contains two items: a folder named "helloExtension" and another folder named "_MACOSX". Since we're looking to install a Chrome extension, the "helloExtension" folder is most likely the extension we need to install. Chrome extensions typically follow a specific folder structure with an "manifest.json" file at the root, and "helloExtension" appears to be a properly named extension folder. Now that I've located what appears to be the extension folder, the next logical step would be to select this folder to proceed with the installation. According to the goal, we need to install a Chrome extension located in the Desktop directory, and we've now found what seems to be that extension.

I have several options at this point: 1. Click on the "helloExtension" folder to select it 2. Click the "Select" button (which would likely select the currently highlighted item) 3. Navigate into the folder to confirm its contents

Since the "helloExtension" folder is the most likely candidate for the extension we need to install, and it's already visible in the file browser, clicking on this folder to select it is the most logical next action. This will highlight the folder and prepare it for selection, after which I can click the "Select" button to complete the installation process.

I expect that after clicking on the "helloExtension" folder, it will become highlighted, indicating it's selected. Then I can proceed to click the "Select" button to load this unpacked extension into Chrome.

**Action:** Click on the "helloExtension" folder in the file browser dialog to select it for installation.

**Code:** pyautogui.click(x=0.454, y=0.456)

- - - - - - - - - - - - - - - - - - - - - - - - - - - - - - - - - -

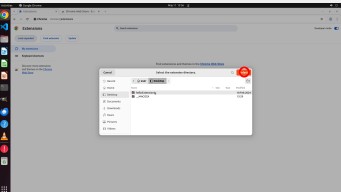

**Step 14**

**Thought:** After clicking on the "helloExtension" folder in the file browser dialog, I can see that the folder is now selected. This is exactly what we needed to do to identify the extension we want to install. The file browser dialog shows we're in the Desktop directory, and we can see the "helloExtension" folder that contains our newly developed Chrome extension.

Now that we've selected the "helloExtension" folder, the next logical step is to confirm this selection by clicking the "Select" button in the file browser dialog. This button is prominently displayed in orange in the top-right corner of the dialog. Clicking this button will finalize our selection and proceed with the installation process.

The installation workflow so far has been: 1. We enabled Developer mode in Chrome 2. We clicked "Load unpacked" to open the file browser 3. We navigated to the Desktop directory 4. We selected the "helloExtension" folder

The final step to complete the installation is to click the "Select" button, which will tell Chrome to load the unpacked extension from the selected folder. After clicking this button, Chrome should install the extension and possibly show a confirmation message or automatically add the extension to the browser. I expect that after clicking the "Select" button, the file browser dialog will close, and we'll return to the Extensions page in Chrome where we might see the newly installed extension appear in the list of installed extensions, or possibly see a notification about the successful installation.

**Action:** Click the "Select" button in the top-right corner of the file browser dialog to confirm the selection of the helloExtension folder and complete the installation process.

**Code:** pyautogui.click(x=0.718, y=0.358)

- - - - - - - - - - - - - - - - - - - - - - - - - - - - - - - - - -

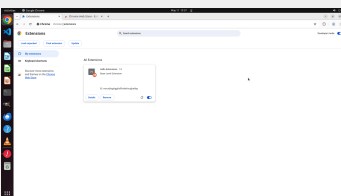

**Step 15**

**Thought:** After clicking the "Select" button in the file browser dialog, the extension installation process has completed successfully. The dialog has closed and we've been returned to the Extensions management page where we can now see that the "Hello Extensions" extension has been installed. The extension appears with its icon, version number (1.0), and description ("Base Level Extension").Looking at our progress toward the goal of installing the Chrome extension: 1. We enabled Developer mode in the Extensions page ✓ 2. We clicked the "Load unpacked" button which opened the file browser dialog ✓ 3. We navigated to the Desktop directory ✓ 4. We selected the "helloExtension" folder ✓ 5. We clicked the "Select" button to confirm the installation ✓ 6. The extension has now been successfully installed ✓ The task has been completed successfully! The extension is now installed and appears in the Extensions management page. I can see that it's enabled (the toggle switch is turned on). If we wanted to configure any specific settings for this extension, we could click on the "Details" button which would likely show more options for the extension.Since our goal was to install the extension manually from the Desktop directory, and that has been accomplished, there's no further action needed for the basic installation task. The extension is now available for use in Chrome.

**Action:** Terminate the process with success status as the extension installation is complete.

**Code:** computer.terminate(status='success')

# H AgentNet Training Data Example

## H.1 L1 Data Example

L1 data includes CoT-L1(Action) in its inner monologue. The assistant's final output will be included in the loss calculation (green color content).



**L1 Data Example**

```
1  [
2      {
3          "role": "system",
4          "content": "You are a GUI agent. You are given a task
               and a screenshot of the screen. You need to perform a
                series of pyautogui actions to complete the task.For
                each step, provide your response in this format:
               Action:  Provide clear, concise, and actionable
               instructions:  - If the action involves interacting
               with a specific target:    - Describe target
               explicitly without using coordinates    - Specify
               element names when possible (use original language if
                non-English)    - Describe features (shape, color,
               position) if name unavailable    - For window control
                buttons, identify correctly (minimize, maximize,
               close)  - if the action involves keyboard actions
               like 'press', 'write', 'hotkey':    - Consolidate
               repetitive keypresses with count    - Specify
               expected text outcome for typing actionsFinally,
               output the action as PyAutoGUI code or the following
               functions:- {"name": "computer.triple_click", "
               description": "Triple click on the screen", "
               parameters": {"type": "object", "properties": {"x": {
               "type": "number", "description": "The x coordinate of
                the triple click"}, "y": {"type": "number", "
               description": "The y coordinate of the triple click"
               }}, "required": ["x", "y"]}}- {"name": "computer.
               terminate", "description": "Terminate the current
               task and report its completion status", "parameters":
                {"type": "object", "properties": {"status": {"type":
                "string", "enum": ["success", "failure"], "
               description": "The status of the task"}}, "required":
                ["status"]}}"
5      },
6      {
7          "role": "assistant",
8          "content": "# Step 1:
9          ## Action:Click on the "Destination Columns-A..."
               document thumbnail to open the document.
10
11         # Step 2:
12         ## Action:Select the word "Travelers" at the beginning
               of the second paragraph by clicking at the start of
               the word and dragging to the end of it."
13     },
14     {
15         "role": "user",
16         "image":
```



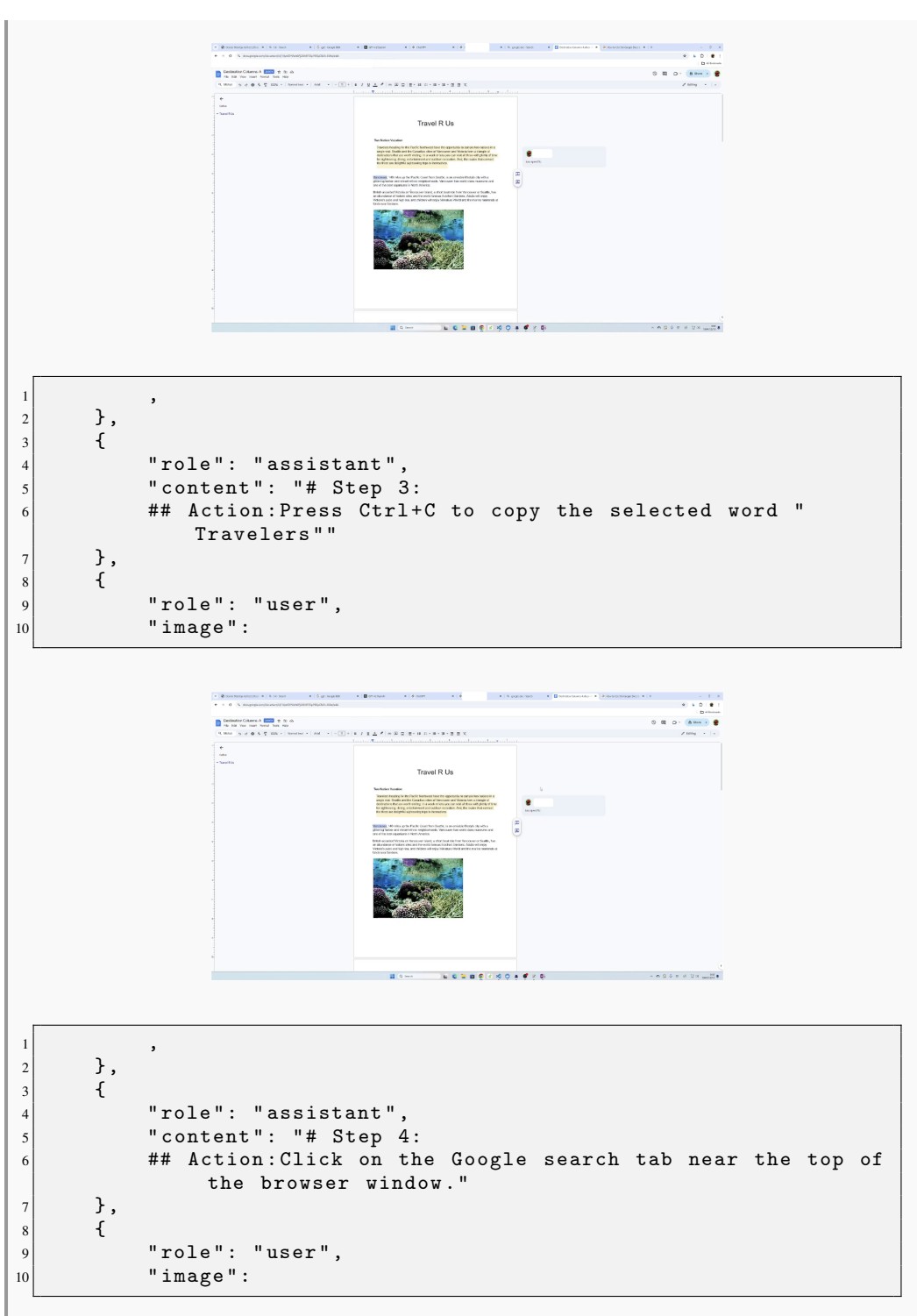

```
1          ,
2      },
3      {
4          "role": "assistant",
5          "content": "# Step 3:
6          ## Action:Press Ctrl+C to copy the selected word "
                Travelers""
7      },
8      {
9          "role": "user",
10         "image":
```

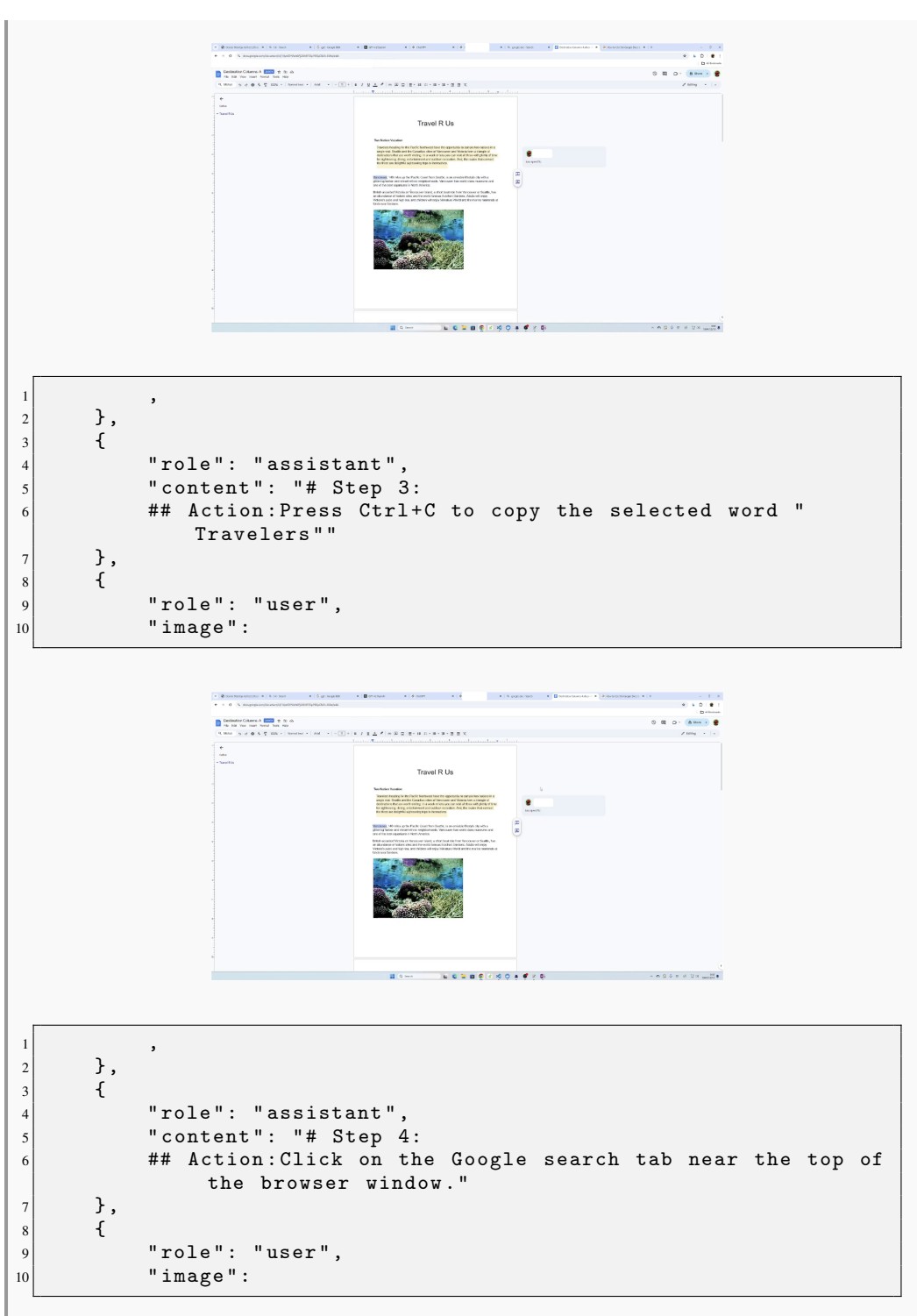

```
1          ,
2      },
3      {
4          "role": "assistant",
5          "content": "# Step 4:
6          ## Action:Click on the Google search tab near the top of
                the browser window."
7      },
8      {
9          "role": "user",
10         "image":
```

```
 1          ,
 2      },
 3      {
 4          "role": "user",
 5          "content": "# Task Instruction:
 6          Could you open the document 'Destination Columns-A',
 7              find the first word of the second paragraph, look up
 8              what that word means on Wikipedia, and then add that
 9              Wikipedia description as a footnote to that word in
10              the document?
11          Please generate the next move according to the
12              screenshot, task instruction and previous steps (if
13              provided)."
14      },
15      {
16          "role": "assistant",
17          "content": "# Step 5:
18          ## Action:Click on the Google search box where "gpt" is
19              currently displayed to prepare to enter a new search
20              query.
21
22          ## Code:'''python\npyautogui.click(x=0.157, y=0.1229)'''
23              "}
24      }
25  ]
```

(Line numbering as shown in source: 1–16)

```
 1          ,
 2      },
 3      {
 4          "role": "user",
 5          "content": "# Task Instruction:
 6          Could you open the document 'Destination Columns-A',
                find the first word of the second paragraph, look up
                what that word means on Wikipedia, and then add that
                Wikipedia description as a footnote to that word in
                the document?
 7          Please generate the next move according to the
                screenshot, task instruction and previous steps (if
                provided)."
 8      },
 9      {
10          "role": "assistant",
11          "content": "# Step 5:
12          ## Action:Click on the Google search box where "gpt" is
                currently displayed to prepare to enter a new search
                query.
13
14          ## Code:'''python\npyautogui.click(x=0.157, y=0.1229)'''
                "}
15      }
16  ]
```

## H.2   L2 Data Example

L2 data includes CoT-L2(Thought, Action) in its inner monologue. The assistant's final output will be included in the loss calculation.

**L2 Data Example**

```
 1
 2  [
 3
 4      {
 5          "role": "system",
 6          "content": "You are a GUI agent. You are given a task
                and a screenshot of the screen. You need to perform a
                series of pyautogui actions to complete the task.For
                each step, provide your response in this format:
                Thought:  - Step by Step Progress Assessment:    -
                Analyze completed task parts and their contribution
                to the overall goal    - Reflect on potential errors,
                unexpected results, or obstacles    - If previous
```

```
                    action was incorrect, predict a logical recovery step
                      - Next Action Analysis:     - List possible next
                    actions based on current state     - Evaluate options
                    considering current state and previous actions     -
                    Propose most logical next action     - Anticipate
                    consequences of the proposed action  - For Text Input
                     Actions:    - Note current cursor position     -
                    Consolidate repetitive actions (specify count for
                    multiple keypresses)    - Describe expected final
                    text outcome  - Use first-person perspective in
                    reasoningAction:  Provide clear, concise, and
                    actionable instructions:  - If the action involves
                    interacting with a specific target:    - Describe
                    target explicitly without using coordinates    -
                    Specify element names when possible (use original
                    language if non-English)    - Describe features (
                    shape, color, position) if name unavailable    - For
                    window control buttons, identify correctly (minimize,
                     maximize, close)  - if the action involves keyboard
                    actions like 'press', 'write', 'hotkey':    -
                    Consolidate repetitive keypresses with count    -
                    Specify expected text outcome for typing
                    actionsFinally, output the action as PyAutoGUI code
                    or the following functions:- {"name": "computer.
                    triple_click", "description": "Triple click on the
                    screen", "parameters": {"type": "object", "properties
                    ": {"x": {"type": "number", "description": "The x
                    coordinate of the triple click"}, "y": {"type": "
                    number", "description": "The y coordinate of the
                    triple click"}}, "required": ["x", "y"]}}- {"name": "
                    computer.terminate", "description": "Terminate the
                    current task and report its completion status", "
                    parameters": {"type": "object", "properties": {"
                    status": {"type": "string", "enum": ["success", "
                    failure"], "description": "The status of the task"}},
                     "required": ["status"]}}"
 6          },
 7          {
 8              "role": "assistant",
 9              "content": "# Step 1:
10              ## Action:Click on the "Destination Columns-A..."
                    document thumbnail to open the document.

11
12              # Step 2:
13              ## Action:Select the word "Travelers" at the beginning
                    of the second paragraph by clicking at the start of
                    the word and dragging to the end of it."
14          },
15          {
16              "role": "user",
17              "image":
```

```
1          ,
2     },
3     {
4         "role": "assistant",
5         "content": "# Step 3:
6         ## Action:Press Ctrl+C to copy the selected word "
                 Travelers""
7     },
8     {
9         "role": "user",
10        "image":
```

```
1          ,
2     },
3     {
4         "role": "assistant",
5         "content": "# Step 4:
6         ## Action:Click on the Google search tab near the top of
                 the browser window."
7     },
8     {
9         "role": "user",
10        "image":
```

```
1            ,
2        },
3        {
4            "role": "user",
5            "content": "# Task Instruction:
6            Could you open the document 'Destination Columns-A',
                find the first word of the second paragraph, look up
                what that word means on Wikipedia, and then add that
                Wikipedia description as a footnote to that word in
                the document?
7            Please generate the next move according to the
                screenshot, task instruction and previous steps (if
                provided)."
8        },
9        {
10            "role": "assistant",
11            "content": "# Step 5:
12            ## Thought:After clicking on the Google search tab, I
                can now see that the browser has successfully
                switched to a Google search page. This is a step
                forward in my task, as I need to search for the word
                "Travelers" on Wikipedia to complete the footnote
                insertion.Currently, I see that the search box
                contains "gpt" rather than the word "Travelers" that
                I copied from the document. To proceed with my task,
                I need to:1. Clear the current search term "gpt" from
                the search box2. Enter "Travelers Wikipedia" or
                navigate directly to Wikipedia and search there3.
                Find relevant information about "Travelers" on
                Wikipedia4. Return to the Google Docs document and
                insert this information as a footnoteMy progress on
                the overall task so far:1. I've opened the "
                Destination Columns-A" document v2. I've identified
                and selected the first word of the second paragraph (
                "Travelers") v3. I've copied this word to the
                clipboard v4. I've switched to a browser tab for
                searching v5. Now I need to clear the current search
                and look up "Travelers" on Wikipedia6. Then I need to
                 insert the Wikipedia description as a
                footnoteLooking at the current state, I need to click
                 directly on the search box where "gpt" is currently
                displayed. By clicking on the search box, I'll be
                able to select all the text, delete it, and replace
                it with "Travelers Wikipedia" to search for
                information about Travelers on Wikipedia. This is the
                 most efficient next action as it allows me to
                directly modify the search query without any
                additional steps.After clicking on the search box, I
```

```
              expect the text cursor to be positioned within the
              box, allowing me to edit the current search term. I
              can then clear "gpt" and type "Travelers Wikipedia"
              to continue with my search. This will help me find
              the Wikipedia page for "Travelers" so I can obtain
              the description needed for the footnote.
13
14        ## Action:Click on the Google search box where "gpt" is
              currently displayed to prepare to enter a new search
              query.
15
16        ## Code:'''python\npyautogui.click(x=0.157, y=0.1229)'''
              "}
17    }
18 ]
```

## H.3  L3 Data Example

L3 data includes CoT-L3(Observation, Thought, Action) in its inner monologue. The assistant's final output will be included in the loss calculation.

L3 Data Example

```
1 [
2     {
3         "role": "system",
4         "content": "You are a GUI agent. You are given a task
              and a screenshot of the screen. You need to perform a
              series of pyautogui actions to complete the task.For
              each step, provide your response in this format:
              Observation:  - Describe the current computer state
              based on the full screenshot in detail.   -
              Application Context:    - The active application    -
              The active window or page    - Overall layout and
              visible interface  - Key Elements:    - Menu items
              and toolbars    - Buttons and controls    - Text
              fields and content    - Dialog boxes or popups    -
              Error messages or notifications    - Loading states
              - Other key elements  - Describe any content,
              elements, options, information or clues that are
              possibly relevant to achieving the task goal,
              including their name, content, or shape (if possible)
              .Thought:  - Step by Step Progress Assessment:    -
              Analyze completed task parts and their contribution
              to the overall goal    - Reflect on potential errors,
              unexpected results, or obstacles    - If previous
              action was incorrect, predict a logical recovery step
              - Next Action Analysis:    - List possible next
              actions based on current state    - Evaluate options
              considering current state and previous actions    -
              Propose most logical next action    - Anticipate
              consequences of the proposed action  - For Text Input
              Actions:    - Note current cursor position    -
              Consolidate repetitive actions (specify count for
              multiple keypresses)    - Describe expected final
              text outcome  - Use first-person perspective in
              reasoningAction:  Provide clear, concise, and
              actionable instructions:  - If the action involves
              interacting with a specific target:    - Describe
              target explicitly without using coordinates    -
```

```
              Specify element names when possible (use original
              language if non-English)    - Describe features (
              shape, color, position) if name unavailable    - For
              window control buttons, identify correctly (minimize,
              maximize, close)  - if the action involves keyboard
              actions like 'press', 'write', 'hotkey':    -
              Consolidate repetitive keypresses with count    -
              Specify expected text outcome for typing
              actionsFinally, output the action as PyAutoGUI code
              or the following functions:- {"name": "computer.
              triple_click", "description": "Triple click on the
              screen", "parameters": {"type": "object", "properties
              ": {"x": {"type": "number", "description": "The x
              coordinate of the triple click"}, "y": {"type": "
              number", "description": "The y coordinate of the
              triple click"}}, "required": ["x", "y"]}}- {"name": "
              computer.terminate", "description": "Terminate the
              current task and report its completion status", "
              parameters": {"type": "object", "properties": {"
              status": {"type": "string", "enum": ["success", "
              failure"], "description": "The status of the task"}},
               "required": ["status"]}}"
5      },
6      {
7          "role": "assistant",
8          "content": "# Step 1:
9          ## Action:Click on the "Destination Columns-A..."
              document thumbnail to open the document.
10
11          # Step 2:
12          ## Action:Select the word "Travelers" at the beginning
              of the second paragraph by clicking at the start of
              the word and dragging to the end of it."
13      },
14      {
15          "role": "user",
16          "image":
```

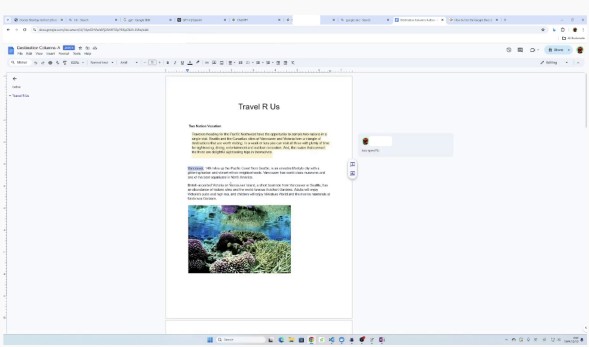

```
1          ,
2      },
3      {
4          "role": "assistant",
5          "content": "# Step 3:
6          ## Action:Press Ctrl+C to copy the selected word "
              Travelers""
7      },
8      {
```

```
9          "role": "user",
10         "image":
```

```
1          ,
2      },
3      {
4          "role": "assistant",
5          "content": "# Step 4:
6          ## Action:Click on the Google search tab near the top of
                  the browser window."
7      },
8      {
9          "role": "user",
10         "image":
```

```
1          ,
2      },
3      {
4          "role": "user",
5          "content": "# Task Instruction:
6          Could you open the document 'Destination Columns-A',
                  find the first word of the second paragraph, look up
                  what that word means on Wikipedia, and then add that
                  Wikipedia description as a footnote to that word in
                  the document?
7          Please generate the next move according to the
                  screenshot, task instruction and previous steps (if
                  provided)."
8      },
9      {
10         "role": "assistant",
```

```
11        "content": "# Step 5:
12      ## Observation:I'm looking at a Google search page where
             "gpt" has been entered in the search box. The
             interface shows Google search results for "gpt" with
             various tabs available at the top (All, Images,
             Videos, Shopping, News, More). The search results
             show information about GPT and related technologies.
             Multiple search results are visible, including:-
             Information about GPT-4- ChatGPT (from https://
             chatgpt.com)- Multiple websites with GPT-related
             content The browser has multiple tabs open at the top
             , including a tab for "Destination Columns-A"
             document, which appears to be a Google Docs document
             that was previously being worked on. There's also a
             search bar at the top with the query "gpt" entered.
             Below the search results are some related questions
             in a FAQ section, such as "What does the GPT stand
             for?" and "Is ChatGPT free?" among others.The Google
             search interface includes various filters and options
             to refine the search, including some buttons like "
             Gpt-4 " (GPT-4 relevance) and several other filter
             options.
13
14      ## Thought:After clicking on the Google search tab, I
             can now see that the browser has successfully
             switched to a Google search page. This is a step
             forward in my task, as I need to search for the word
             "Travelers" on Wikipedia to complete the footnote
             insertion.Currently, I see that the search box
             contains "gpt" rather than the word "Travelers" that
             I copied from the document. To proceed with my task,
             I need to:1. Clear the current search term "gpt" from
             the search box2. Enter "Travelers Wikipedia" or
             navigate directly to Wikipedia and search there3.
             Find relevant information about "Travelers" on
             Wikipedia4. Return to the Google Docs document and
             insert this information as a footnoteMy progress on
             the overall task so far:1. I've opened the "
             Destination Columns-A" document v2. I've identified
             and selected the first word of the second paragraph (
             "Travelers") v3. I've copied this word to the
             clipboard v4. I've switched to a browser tab for
             searching v5. Now I need to clear the current search
             and look up "Travelers" on Wikipedia6. Then I need to
             insert the Wikipedia description as a
             footnoteLooking at the current state, I need to click
             directly on the search box where "gpt" is currently
             displayed. By clicking on the search box, I'll be
             able to select all the text, delete it, and replace
             it with "Travelers Wikipedia" to search for
             information about Travelers on Wikipedia. This is the
             most efficient next action as it allows me to
             directly modify the search query without any
             additional steps.After clicking on the search box, I
             expect the text cursor to be positioned within the
             box, allowing me to edit the current search term. I
             can then clear "gpt" and type "Travelers Wikipedia"
             to continue with my search. This will help me find
             the Wikipedia page for "Travelers" so I can obtain
             the description needed for the footnote.
15
```

```
16          ## Action:Click on the Google search box where "gpt" is
                currently displayed to prepare to enter a new search
                query.
17
18          ## Code:'''python\npyautogui.click(x=0.157, y=0.1229)'''
                "}
19      }
20 ]
```

