# OpenReview forum: "OpenCUA: Open Foundations for Computer-Use Agents"
_NeurIPS.cc/2025/Conference — NeurIPS 2025 spotlight_

### Official Review · Reviewer_8D4Y · 2025-06-19

**Clarity:** 4
**Significance:** 2
**Originality:** 3
**Rating:** 6
**Confidence:** 5

**Summary:**

This study introduces AgentNet, an open-source framework designed to capture human-computer user interactions, curate a rich multi-modal dataset, and train agents to automate user tasks through a long Chain-of-Thought (CoT) reasoning. The framework fine-tunes off-the-shelf vision-language models using supervised learning on (state, action) pairs derived from user interaction data. Given the growing demand for automation and intelligent agents, this work fills a critical gap by providing a free, open-source, and non-proprietary solution for developing and evaluating Computer Use Agents (CUAs).

I find this line of research innovative and clear. However, I have some concerns with the reasoning method used here and the results shown in Tables 2 and 3. I will express those concerns later in this review.

**Questions:**

1. Did you observe any conditions under which the agent would get stuck in a loop or fail to terminate properly? If so, it would be valuable to document these cases. Since this is an open-source framework, highlighting such limitations would benefit the research community by guiding future efforts to develop more robust and advanced methods.
2. Recent researches on various in-context learning methods and prompting have discovered that CoT often leads to hallucinations in LLMs. To overcome this, the reflector module is added; however, there are some advanced methods, such as REACT: Synergizing Reasoning and Acting in Language Models, Toolformer (connecting APIs to LLMs), LLM planning with Monte-Carlo Tree search, etc. These methods have shown to reduce or eliminate hallucinations and perform better than the CoT method. I advise the authors to try at least one other reasoning method apart from CoT and check for gains.

**Ethical Concerns:**

["NO or VERY MINOR ethics concerns only"]

**Final Justification:**

I am convinced that the authors' proposed method is a valuable contribution to the field of Agentic-AI. Their well-curated multi-stage pipeline will offer many research directions in the future for building autonomous agents in different work domains.

I found this rebuttal phase very useful in helping the authors identify the limitations of their work, which I strongly suggest that they should add in the final version. I was also able to understand their long-term vision with this research line, and therefore I after being fully convinced of their methods, I have decided to increase my score.

**Limitations:**

The authors have clearly stated all the limitations of this study in the Appendix.

**Paper Formatting Concerns:**

No concerns.

**Quality:**

3

**Strengths And Weaknesses:**

Strengths:

1. The main objective of this study is to provide a free, open-source framework to advance research in agentic development.
2. The framework supports multi-modality, i.e., language and vision.
3. The authors have meticulously designed an end-to-end pipeline, also shown in the great illustration in Figure 1.
4. The dataset collected using this pipeline is rich, and I appreciate that the authors have carefully compared their dataset with other existing datasets in Table 1 and Figure 3.
5. The proposed framework not only surpasses 'some' state-of-the-art models in performance but also offers an open-source tool for the scientific community. I use the word 'some' deliberately, as I have concerns about the results shown in Tables 2 and 3, which I will address below in the 'Weakness' section.
6. The authors have designed a reflector module, which I find an extremely important addition to this research.

Weakness:

1. [MINOR] Missing capitalization after period in Line 201. Figure 8 comes after Figure 7. Typo in Line 134. Please fix these very minor issues.
2. I recommend that authors check and consider citing the paper 'Camel: Communicative Agents for "Mind" Exploration of Large Language Model Society. The paper is very similar to this work and also offers an open-source agentic framework to the research community.
3. I have deducted one point due to concerns regarding the results presented in Table 2. The authors claim that fine-tuning alone leads to poor performance and that incorporating long Chain-of-Thought (CoT) reasoning significantly improves outcomes. However, the reported results for OSWorld (15 steps) appear unconvincing when compared to other models. While the proposed framework outperforms several baselines, it does not surpass comparable architectures such as OpenAI CUA and UI-TARS variants. If the authors believe this performance gap is due to factors such as model architecture size, training data scale, or a combination of both, they should clearly articulate this to support their claim.
Furthermore, in the case of OSWorld (50 steps), the proposed framework still underperforms relative to existing methods. The comparison for the WAA task is also insufficient to determine whether the framework offers a significant improvement over current approaches. Therefore, I encourage the authors to provide additional analysis explaining the performance limitations of their framework on OSWorld, and to clarify the conditions under which their method is most effective. Do you think that the model under performs in case of 50 steps due to hallucination which is common in the CoT reasoning method?

If the main focus of the paper is on the dataset pipeline, then the authors should clearly mention that their objective is not to compete with state-of-the-art models. In that case, the weak results in table 2 would be sufficient for this study.

Finally, Table 3 results are stronger than Table 2 results and significantly outperform existing models.

---

> ### Author Rebuttal · Authors · 2025-07-31
>
> We sincerely thank reviewer 8D4Y for the insightful questions and suggestions. We are grateful that you find our work “innovative and clear,” and we appreciate the recognition of the contributions our framework, dataset, and models make to the agent-research community.
> ## Response to Weakness 1:
> We apologize for the typos and thank the reviewer for the careful reading. All errors will be fixed in the next revision.
>
> ## Response to Weakness 2:
> We thank the reviewer for pointing us to “Camel: Communicative Agents for ‘Mind’ Exploration of Large Language Model Society.” This open-source framework, together with its accompanying library and datasets, has substantially advanced multi-agent research. We will cite this related work and expand our discussion of multi-agent approaches in our next revision.
>
> ## Response to Weakness 3 and Question 1:
> We thank the reviewer for the thoughtful feedback on our model’s performance and limitations.
>
> (1) Over the last two months, we have scaled our data to 40K trajectories and added additional grounding data and scale our model from 7B to 32B size. We fine-tuned Qwen2.5-VL-7B [1] and Qwen2.5-VL-32B[1] on these datasets. As shown in the following table, our 32B model achieves **34.8%** (maximum score **35.7%**), surpassing UI-TARS-72B-DPO= and even GPT-4o based OpenAI CUA, although they have much more parameters. Our model is now a new state-of-the-art among the open-source models. We will include these new results in the next revision and release both models
>
> | Model | Size | OSWorld | ScreenSpot-Pro |
> | --- | ---  | ---     | ---            |
> | UI-TARS-72B-DPO | 72 B | 27.1 %  | 39.4 %         |
> | OpenAI CUA (GPT-4o) | —  | 31.4 %  | —              |
> | AgentNet-Qwen2.5-VL-7B  | 7 B  | 27.9 %  | 50.0 %         |
> | AgentNet-Qwen2.5-VL-32B | 32 B | 34.8 %  | 55.3 %         |
>
> (2) We thank the reviewer for the suggestion for adding more limitations and error analysis. We have inspected the evaluation results on OSWorld and summarize the errors into the following categories:
>
> 1. Insufficient task knowledge: Foundation models still lack domain-specific GUI knowledge. Each application has unique UI conventions and operation logic; without explicit exposure, both agents and even humans can struggle.
>    Example: In the task “I have a lookup table for the officers of each branch. Please fill the second table using VLOOKUP.” the agent does not know the VLOOKUP function and therefore fails.
>
> 2. High-precision grounding errors: Tasks that require pixel-accurate actions frequently fail.
>    Example: In “Change the 2 in ‘H₂O’ to a subscript,” the agent must precisely drag-select only the “2,” but it often selects more characters.
>
> 3. Action repetition and failure to terminate: When the agent misjudges a state after an action, it may keep executing (and reflecting on) the same incorrect action. Occasionally it realizes and does a different action to break the loop, but sometimes it loops indefinitely. In some cases, the agent fails to notice that the task is already complete and keeps acting instead of terminating.
>
> 4. Long horizon tasks: OSWorld contains tasks requiring > 30-50 gold actions. Maintaining context over so many steps remains challenging.
>    Example: “Organize my desktop by placing academic papers in ‘Paper_reading’, coding projects in ‘Projects’, and everything else in ‘Miscellaneous’. For files without clear names, determine the category by content.” It requires take actions on many files and folders.
>
> 5. Error perception: Although the agent model has a certain level of ability to identify errors and reflect, in some cases that requires high accuracy, it is still challenging. In fine-grained text-editing tasks, the agent may insert a word one character off, yet judge the action correct and proceed.
>
> (3) During the error analysis, we notice that a small portion of steps contains hallucinations in their CoT. The hallucination mainly includes planning and predicted action misalignment and misinterpreting the success of the last action. To study the source of this hallucination in the evaluation, we inspected the CoT of our training data. We use GPT-4 to check each step’s CoT for these kinds of errors in our data. We find that around 3% of trajectories contain one or two such errors. We think that this hallucination indeed has a negative impact on the performance, it may cause incorrect action prediction and partially cause the action repetition. Although hallucination exists in our training data, our experiments show that as long as the hallucination rate is kept within an acceptable low percent, scaling the dataset continues to yield performance gains. (After identifying the errors in the CoT, we are now re-synthesize the CoTs of these actions. We appreciate the reviewer’s insightful question.)
> ## Response to Question 2:
> We thank the reviewers suggestion on investigating in-context learning and prompting-based methods.
>
> To better illustrate the following new experiment results and discussion, we would like first to elaborate more about our method. The motivation of our work is to facilitate the research of agentic foundation models, which have the ability to act as an agent themselves and automatically perceive environment, plan, reflect, reason and take actions iteratively. Claude 3.7 Sonnet [2], Claude 4 Sonnet [3], and Kimi K2 [4] fall in this line of work. They also have the ability to be part of the multi-agent systems through prompting thanks to their generalization capability and the agentic data in the pre-training. To integrate the required agentic abilities, including planning, reflection and memory, into the foundation model, we created a pipeline to first build a trajectory dataset. Each step of the trajectory has observation, meticulously designed reflective long CoT and grounded action. Base on ActRe [5], we designed a pipeline including a reflector and a generator to synthesize these CoTs for training. After training, the model's computer environment understanding and agentic capablities are improved, so the foundation model can reason first and output the action in the same prediction. During executing, the model follows the ReAct [6] style: observation-> (thought, action)->observation… In the ‘thought’, the model may make plans and reflect the former errors.
>
> There are several in-context learning methods in the CUA field. For example, Agent S2 [7] carefully designed an agent framework including manager worker and mixture of grounding modules based on Claude 3.7 Sonnet. Based on the framework they improved the performance on OSWorld from 26.0% to 34.5%.  In another work, “Scaling Computer-Use Grounding via User Interface Decomposition and Synthesis” [8], they trained a specialized grounding model called Jedi. The grounding model itself can not complete tasks in the OSWorld, while GPT-4o has general reasoning and GUI knowledge, but without grounding capability, it only has a success rate of 5% on OSWorld. But when using GPT-4o as planner and Jedi as grounder, the OSWorld success rate boosts to 27.0%.
>
> Due to the limit of rebuttal time, we implemented Jedi’s pipeline in our setting and use our own model as planner and grounder. In each step, we first prompt the agent to reason and plan, then we parse the low-level-instruction and give it to the grounder to predict the action that will be executed. We did an experiment on our 32B model and the result is 23.5% in 15 steps and 24.2% in 100 steps. We think the reason Jedi+GPT-4o improves performance, while ours not is because GPT-4o is a good planner but bottlenecked by the grounding capability, but our model has integrated both ability into our model and it naturally doing a ReAct style prediction, splitting it into two stages may affect its internal logic. However, we think a multi-agent framework is still promising by adding another advanced model as a reflector/judger after our model makes predictions to inspect the errors/hallucinations/incorrect screenshot understanding, and we believe this can mitigate the negative effect of repetition and terminate failures.
>
> We also discussed building an MCTS based agent framework. We think that the most challenging part in the system is exploring the environment and assessing the reward, because the computer environment can not be backtraced in many cases, for example, many changes in the live environment can not be reverted. Building such a system would require a reward model in the CUA domain and a digital world model or a strong engineering improvement to overcome this barrier. We will explore this direction in our future works.
>
> Thanks again to the reviewer for the insightful questions and suggestions!
>
> [1] Bai, Shuai, Keqin Chen, Xuejing Liu, Jialin Wang, Wenbin Ge, Sibo Song, Kai Dang et al. "Qwen2. 5-vl technical report." arXiv preprint arXiv:2502.13923 (2025).
>
> [2] Anthropic. Claude’s extended thinking.
>
> [3] Anthropic. Introducing claude 4, 5 2025.
>
> [4] Team, Kimi, Yifan Bai, Yiping Bao, Guanduo Chen, Jiahao Chen, Ningxin Chen, Ruijue Chen et al. "Kimi K2: Open Agentic Intelligence." arXiv preprint arXiv:2507.20534 (2025).
>
> [5] Yang, Zonghan, Peng Li, Ming Yan, Ji Zhang, Fei Huang, and Yang Liu. "React meets actre: When language agents enjoy training data autonomy." arXiv preprint arXiv:2403.14589 (2024).
>
> [6] Yao, Shunyu, Jeffrey Zhao, Dian Yu, Nan Du, Izhak Shafran, Karthik Narasimhan, and Yuan Cao. "React: Synergizing reasoning and acting in language models." In International Conference on Learning Representations (ICLR). 2023.
>
> [7] Agashe, Saaket, Kyle Wong, Vincent Tu, Jiachen Yang, Ang Li, and Xin Eric Wang. "Agent s2: A compositional generalist-specialist framework for computer use agents." arXiv preprint arXiv:2504.00906 (2025).
>
> [8] Xie, Tianbao, Jiaqi Deng, Xiaochuan Li, Junlin Yang, Haoyuan Wu, Jixuan Chen, Wenjing Hu et al. "Scaling Computer-Use Grounding via User Interface Decomposition and Synthesis." arXiv preprint arXiv:2505.13227 (2025).

---

> ### Comment · Reviewer_8D4Y · 2025-08-01
> **Response to rebuttal**
>
> Thank you, authors, for addressing each one of my queries. Below is my response to your points.
>
> **1. Addition of new models:** Thanks for adding these results. Yes, these are much needed and will strengthen your claims.
>
> **2. Limitation section:** I appreciate that the authors have categorized their limitations into individual categories. This precisely indicates where the agent currently lacks. This section alone deserves one point. I strongly encourage the authors to add all these details, including their points on hallucinations with CoT in their manuscript (S.I section), to better help the community of agentic-AI.
>
> **3. Pipeline explanation:** I find the authors' explanation on their choice of method very informative and agree with their points.
>
> I find your points convincing, and I believe that your contribution will advance the agentic-AI field. Therefore, after being fully convinced that your methods are correct, I will increase my score.

---

> > ### Author Response · Authors · 2025-08-02
> >
> > We sincerely thank Reviewer 8D4Y for the thoughtful follow-up and for considering an increased score.
> > The discussions and feedbacks are extremely helpful for us. We will incorporate all points in our next revision, especially the expanded error-analysis section and the discussion of hallucinations in CoT. These additions will make the paper clearer and more useful to the agentic-AI community. Thank you again for your constructive engagement!

---

### Official Review · Reviewer_q328 · 2025-06-23

**Clarity:** 3
**Significance:** 3
**Originality:** 3
**Rating:** 5
**Confidence:** 3

**Summary:**

This paper introduces AGENTNET, an open-source framework for training and evaluating computer-use agents (CUAs). It consists of several key components, including a cross-platform annotation tool, a large-scale dataset of 27.8K human-annotated desktop task trajectories, reasoning-augmented training with multi-level Chain-of-Thought, and an offline benchmark (AGENTNETBENCH) to enable scalable evaluation. Two fine-tuned vision-language models (Qwen2-VL-7B, Kimi-VL-A3B) are presented and achieve superior performance among open-source 7B models on OSWorld and WindowsAgentArena. All components of the proposed framework including tool, data, models, benchmark will be released, which offers an open foundation for future CUA research.

**Questions:**

- How well does the dataset transfer to novel domains or unseen applications?
- Are there any plans to adapt AGENTNET for online interaction settings?

**Ethical Concerns:**

["NO or VERY MINOR ethics concerns only"]

**Final Justification:**

The author's rebuttal addressed the minor concerns I had. I decide to maintain my original positive ratings.

**Limitations:**

Yes

**Quality:**

3

**Strengths And Weaknesses:**

Strengths:
- The proposed dataset, tools, and benchmarks fill a major transparency gap in current CUA research and enable reproducibility.
- The proposed dataset covers real-world workflows across Windows, macOS, and Ubuntu, with realistic errors and long-horizon tasks.
- The multi-level reflective CoT framework is thoughtful and empirically validated through ablation studies.

Weaknesses:
- Despite the scale and quality of the dataset, AgentNet-7B shows only marginal improvement over prior 7B models (e.g., UI-TARS-7B-SFT) on OSWorld, raising questions about generalization benefits.
- The proposed dataset heavily features Windows/macOS environments, while OSWorld focuses on Ubuntu tasks, which may limit out-of-domain performance.
- The proposed benchmark is offline only, which may not fully reflect performance in realistic CUA deployment settings.

---

> ### Author Rebuttal · Authors · 2025-07-31
>
> We sincerely thank Reviewer q328 for recognizing AgentNet’s contributions—particularly our realistic, diverse dataset and reflective CoT framework. Over the past two months we have further expanded both data and models. We now have 40 K trajectories with additional grounding data, and we have fine-tuned Qwen 2.5-VL-7B [1] and Qwen 2.5-VL-32B [1] on this corpus. The 7B model achieves a 27.9 % success rate on OSWorld, while the 32 B model reaches **34.8 %**—state-of-the-art among open-source systems and surpassing GPT-4o based OpenAI CUA [4]. Both models also excel at grounding: the 32 B version scores **55.3** on ScreenSpot-Pro[2], surpassing the previous SOTA UI-TARS-72B (39.4) [3]. We will incorporate these new results and model checkpoints in the next revision of the paper.
>
> | Model | Size | OSWorld | ScreenSpot-Pro |
> | --- | ---  | ---     | ---            |
> | UI-TARS-72B-DPO | 72 B | 27.1 %  | 39.4 %         |
> | OpenAI CUA (GPT-4o) | —  | 31.4 %  | —              |
> | AgentNet-Qwen2.5-VL 7B  | 7 B  | 27.9 %  | 50.0 %         |
> | AgentNet-Qwen2.5-VL 32B | 32 B | 34.8 %  | 55.3 %         |
>
> ## Response to Weakness 1:
> Through continued data scaling, our model now achieves an average success rate of **34.8 %** on OSWorld, with a highest score of **35.7 %**. This surpasses all open-source baselines by a wide margin and proprietary model OpenAI CUA (GPT-4o).
>
> ## Response to Weakness 2:
> Thank you for the question about dataset domain effects and model generalization. We have also collected tasks on Ubuntu, and our tool supports that platform.
>
> In Figure 5 we analyze system-specific training. We fine-tune Qwen 2-VL-7B on (i) Windows & macOS data and (ii) Ubuntu data, then evaluate on three benchmarks: OSWorld (Ubuntu), WindowsAgentArena (Windows), and our offline benchmark (Windows & macOS). Training on a single operating system yields the highest performance on that same system, yet still transfers some gains to the others.
>
> We additionally evaluate our 32 B model on WindowsAgentArena; it attains an average score of 28.0 (max 30.0). (Manual inspection shows some false negatives caused by environment errors, so the true score is likely even higher.)
>
> ## Response to Weakness 3:
> We fully acknowledge the reviewer’s concern: an offline benchmark cannot mirror every aspect of real-world deployment. We nonetheless chose to build an offline suite—rather than another online benchmark like OSWorld—for three practical reasons:
>
> 1. Live environments are highly dynamic, which makes experiment signals unstable. During extended experiments we observed that a single Chrome update could deprecate UI elements and invalidate previous evaluation metrics. Additional variables—login prompts, CAPTCHA checks, fluctuating network latency—may render results collected just two weeks apart incomparable. An offline setting avoids these shifting sands.
>
> 2. To mitigate the usual limitations of offline testing, we carefully selected representative tasks from our dataset and manually annotated each step. We drew bounding boxes for every click action, and found multiple feasible actions in each state, to capture the full space of valid solutions. This manual curation increases robustness and reduces the chance of false negatives when agents choose alternative but correct strategies.
>
> 3. Because no live environment needs to be spun up, evaluation is fast and reproducible. We can score every training checkpoint, identify the best-performing models, and then proceed to costlier online testing with higher confidence.
>
> ## Response to Question 1:
> Thank you for highlighting the domain-transfer issue. As we discussed in the response to Weakness 2, in Section 5 Figure 5, we show the effect of domain shift. Our conclusion is that:
> 1. There are cross-domain gains. Many low-level GUI patterns—window management, menu traversal, standard shortcuts, drag-and-drop, etc.—are shared across operating systems, applications, and even websites. Training on a mix of domains therefore improves overall robustness.
>
> 2. Tasks requiring specific domain knowledge are hard to be completed only by domain transferring. When a task hinges on domain-specific knowledge—e.g., triggering an uncommon Excel formula or operating a professional CAD tool—generic GUI skills alone are not enough. The agent needs additional task-specific demonstrations or fine-tuning.
>
> ## Response to Question 2:
> For the reasons outlined in our reply to Question 1, we initially designed AgentNetBench as an offline benchmark. Nevertheless, we fully agree that online evaluation is essential, and we have begun experimenting in that direction. The difficulty of building an online evaluation benchmark lies in the evaluation method. OSWorld’s current protocol relies on hand-written evaluation scripts for open-source applications. Extending this approach to a broader range of real-world, closed-source, or rapidly evolving applications is often infeasible, because reliable scripts either do not exist or would require prohibitive maintenance.
> Currently we did some attempts to use a general foundation model to evaluate the agent’s result. We replaced OSWorld’s official scripts with two large language models—GPT-4o and OpenAI o3-mini—and measured their agreement with the ground-truth scripts. .
>
> | Evaluator | Accuracy | F1 score |
> | ---       | ---      | --- |
> | GPT-4o    | 78 %     | 0.71 |
> | o3-mini   | 85 %     | 0.78 |
>
> We are investigating hybrid schemes that combine lightweight domain-specific checks with VLM-based reasoning and try to build an online extension of AgentNetBench supporting diverse, real-life tasks. (We have interpreted “online interaction settings” as referring to an online evaluation benchmark. If that is not what you meant, please let us know so we can address your concern directly.)
>
> [1] Bai, Shuai, Keqin Chen, Xuejing Liu, Jialin Wang, Wenbin Ge, Sibo Song, Kai Dang et al. "Qwen2. 5-vl technical report." arXiv preprint arXiv:2502.13923 (2025).
>
> [2] Li, Kaixin, Ziyang Meng, Hongzhan Lin, Ziyang Luo, Yuchen Tian, Jing Ma, Zhiyong Huang, and Tat-Seng Chua. "Screenspot-pro: Gui grounding for professional high-resolution computer use." arXiv preprint arXiv:2504.07981 (2025).
>
> [3] Qin, Yujia, Yining Ye, Junjie Fang, Haoming Wang, Shihao Liang, Shizuo Tian, Junda Zhang et al. "Ui-tars: Pioneering automated gui interaction with native agents." arXiv preprint arXiv:2501.12326 (2025).
>
> [4] OpenAI. Operator, 2025.

---

> > ### Comment · Reviewer_q328 · 2025-08-04
> >
> > Thank the authors for the detailed response. I decide to maintain my positive ratings.

---

> > > ### Author Response · Authors · 2025-08-04
> > >
> > > We sincerely thank Reviewer q328 for the reply. The questions and feedbacks are very helpful for us. Thank you again for your constructive engagement and the acknowledgement of AgentNet!

---

### Official Review · Reviewer_jJf2 · 2025-06-28

**Clarity:** 3
**Significance:** 4
**Originality:** 2
**Rating:** 5
**Confidence:** 3

**Summary:**

This paper presents an open-source data collection system for CUAs called AgentNet Tool, and releases a dataset collected using AgentNet that is large enough to train CUAs called AgentNet Dataset. Two CUAs are trained on AgentNet Dataset with different levels of reasoning trajectories, as well as grounding and general supervised fine-tuning data. A new offline benchmark, AgentNet Bench, is constructed from the collected data and used for evaluation. Extensive evaluations covering various aspects of the models, including online and offline performances as well as reasoning capabilities, are performed with detailed analyses, and show that the models trained on the proposed dataset outperform the baselines.

**Questions:**

- (typo) On  line 196 “A” should not be capitalized?
- Were there any architectural considerations you had during your research?

**Ethical Concerns:**

["NO or VERY MINOR ethics concerns only"]

**Final Justification:**

I believe this paper lays an important foundation for research in CUAs, which currently lacks a scalable open-source data collection system as well as a large training dataset. Furthermore, the evaluation experiments conducted in this paper as well as the analyses will be a good reference for other researchers to compare different CUAs in a rigorous and fair manner. My comment about the relatively small architectural contributions was a minor one, and the SOTA results the authors provided to me during the discussion period partially addressed it. Taken together, I believe my original score of 5 (Accept) is appropriate.

**Limitations:**

Yes

**Quality:**

3

**Strengths And Weaknesses:**

Strengths
- It presents an open-source data collection system for CUAs, which would contribute greatly to the research community.
- It presents a dataset large enough to be used to train CUAs, another huge win for the wider research community.
- The training recipe for the proposed model covers many different considerations and is backed by reasonable justifications, especially the CoT reasoning dataset.
- The designs of the evaluation experiments are sound and extensive.
- Analyses of the evaluation results are insightful.

Weaknesses
- (minor) The paper’s contribution on model architecture is light.

---

> ### Author Rebuttal · Authors · 2025-07-31
>
> We sincerely thank Reviewer jJf2 for recognizing AgentNet’s contributions. Our goal is to narrow the gap between proprietary and open-source CUA models by delivering the necessary infrastructure, data, models, and benchmarks to the community.
> Over the past two months we have scaled both data and models further. The dataset now contains 40 K trajectories with additional grounding data, and we fine-tuned Qwen 2.5-VL-7B [3] and Qwen 2.5-VL-32B [3] on these datasets. The 7B model reaches a 27.9 % success rate on OSWorld, while the 32B model achieves an average success rate of **34.8 %**—state of the art among open-source agents and surpasses open-sourced UI-TARS-72B-DPO [4]  and proprietary GPT-4o-based OpenAI CUA [5]. Both models also excel at grounding: the 32 B version scores **55.3%** on ScreenSpot-Pro, surpassing the previous SOTA UI-TARS-72B (39.4%).
> | Model | Size | OSWorld | ScreenSpot-Pro |
> | --- | ---  | ---     | ---            |
> | UI-TARS-72B-DPO | 72 B | 27.1 %  | 39.4 %         |
> | OpenAI CUA (GPT-4o) | —  | 31.4 %  | —              |
> | AgentNet-Qwen2.5-VL-7B  | 7 B  | 27.9 %  | 50.0 %         |
> | AgentNet-Qwen2.5-VL-32B | 32 B | **34.8 %**  | **55.3 %**         |
>
> ## Response to Weakness and Question2:
> We thank the reviewer for asking the model architecture consideration. For open-source consideration, we apply our method on the currently popular open-source models and didn’t make novel improvement on the model structures. But we investigated the effect of model structures and sizes by training on 4 different models including Kimi-VL-A3B[1], Qwen2-VL-7B[2], Qwen2.5-VL-7B[3] and Qwen2.5-VL-32B[3]. Kimi-VL-A3B employs a MoE design, whereas the others are dense models. On the same data scale, the MoE model, despite having more total parameters, underperforms the dense models because fewer parameters are activated. We also observe that Qwen 2.5-VL benefits from a much higher max-pixels budget (12,845,056 versus 829,440 for Qwen2-VL and Kimi-VL-A3B), leading to superior results on high-resolution ScreenSpot-Pro. We appreciate the reviewer’s suggestion and plan to explore more efficient and effective architectures for digital agents in future work.
>
> [1] Team, Kimi, Angang Du, Bohong Yin, Bowei Xing, Bowen Qu, Bowen Wang, Cheng Chen et al. "Kimi-vl technical report." arXiv preprint arXiv:2504.07491 (2025).
>
> [2] Wang, Peng, Shuai Bai, Sinan Tan, Shijie Wang, Zhihao Fan, Jinze Bai, Keqin Chen et al. "Qwen2-vl: Enhancing vision-language model's perception of the world at any resolution." arXiv preprint arXiv:2409.12191 (2024).
>
> [3] Bai, Shuai, Keqin Chen, Xuejing Liu, Jialin Wang, Wenbin Ge, Sibo Song, Kai Dang et al. "Qwen2. 5-vl technical report." arXiv preprint arXiv:2502.13923 (2025).
>
> [4] Qin, Yujia, Yining Ye, Junjie Fang, Haoming Wang, Shihao Liang, Shizuo Tian, Junda Zhang et al. "Ui-tars: Pioneering automated gui interaction with native agents." arXiv preprint arXiv:2501.12326 (2025).
>
> [5] OpenAI. Operator, 2025.
>
> ## Response to Question1:
> We thank the reviewer for the careful reading. Because of length constraints we had to merge several paragraphs in the current draft. In the next version, we will promote “Mixture of grounding 196 and general SFT data” to an independent subsection heading; the sentence beginning “A general-purpose computer-use agent …” will then appear as a full sentence.

---

> > ### Comment · Reviewer_jJf2 · 2025-08-02
> >
> > Thank you for your rebuttal! It's great to hear that you have achieved SOTA results by fine-tuning Qwen2.5-VL-32B on your dataset. My concern about lacking architecture contributions was a minor point, and I'm sure your work will enable future research in that regard. Furthermore, I already gave a high score of 5 (Accept), so I don't think there's a need for me to change my score. Great work!

---

> > > ### Author Response · Authors · 2025-08-03
> > >
> > > Thank you very much for your kind and encouraging response! We truly appreciate your positive evaluation and your thoughtful comments. We hope our work can serve as a foundation for future research, including architectural innovations. Thank you again for your support and your high score—it means a lot to us!

---

### Official Review · Reviewer_6mQE · 2025-07-02

**Clarity:** 3
**Significance:** 2
**Originality:** 3
**Rating:** 5
**Confidence:** 4

**Summary:**

The paper proposes a framework for creating datasets and training models for computer-use agent (CUA). The framework consists of 1) an annotation tool that records video, actions and accessibility tree when human performs computer tasks, 2) a dataset of 27k computer use examples across different OS and applications (a training set and a held-out set), 3) a pipeline that converts the dataset into state-action examples with chain-of-thoughts reasoning, and 4) a model training recipe for CUA. The authors train two models (3B and 7B) on the collected dataset combined with public datasets, and demonstrated good performance on online benchmarks OSWorld and WindowsAgentArena.

**Questions:**

Please see the Weaknesses section above.

**Ethical Concerns:**

["NO or VERY MINOR ethics concerns only"]

**Final Justification:**

I find the data statistics and error analysis provide a lot of useful insights into the dataset and the agent. The SOTA results further prove the usefulness of the dataset, which did take a lot of efforts to create based on the cost details provided by the authors. I think the dataset and the extensive experiments are the main contribution, although authors provided some more argument to differentiate their agent design from previous works like AGUVIS and UI-Tars, but I found those to be minor points.

**Limitations:**

I did not find the discussion of limitations and potential negative societal impact of the work in the paper.

**Quality:**

2

**Strengths And Weaknesses:**

Strengths:
1. The collected dataset covers a wide range of operating systems, websites and applications, with diverse screen sizes and fairly complex tasks (17.2 steps on average). As the first dataset of desktop trajectories, it would facilitate the research in this domain along with the annotation / data conversion tool, which can be used to collect new data.
2. The experiments and analysis show that the dataset is useful for training models that obtain good performance on live CUA benchmarks.

Weaknesses:
1. Other than the new dataset, the novelty of other components is limited as no significantly new methods are proposed. For example, the state-action pair data, the CoT reasoning, and error reflection were already used in previous works like UI-Tars and AGUVIS.
2. Some important details of the dataset are missing. 1) How are the tasks selected? Are they pre-compiled or created by the annotators? 2) How much time and cost needed for creating the dataset? 3) What are the action types and distributions? Any difference across operating systems and/or applications?
3. Including errors and recovery in the trajectory makes sense, but no details are given how they are detected and used. The paper points to Appendix E, but I don't see how error steps are being detected and used in Figure 13. Figure 15 gave an example of reflective CoT, but how this is constructed is not clear (e.g., how can we detect the errors in the trajectories?)
4. There seems no error analysis of the results on the live benchmark. For example, what are the errors for the tasks on which the agent sometimes failed (pass@k < k) or always failed (pass@k = 0)?

---

> ### Author Rebuttal · Authors · 2025-07-31
>
> We sincerely thank Reviewer 6mQE for recognizing both the diversity of our computer-use task dataset and its contribution to the research community, as well as our model’s strong performance on it. We also appreciate the reviewer’s valuable questions and discussion points; we address them below and will add these discussions into the paper.
> ## Response to Weakness 1:
> Although Aguvis[1], UI-TARS[2], and our work all employ structured CoT, our design is more fine-grained and exhibits higher reasoning and reflection quality. Specifically, we introduce a new pipeline that iteratively synthesizes and refines reflective long-form CoT for human demonstrations. Our CoT explicitly covers reflection, planning, memory, and prediction—all important for agent design.
>
> (1) Ours provides richer information than Augivs. In Aguvis Appendix C.1, Fig. 5, the 'reasoning' part is less informative, and in the left example, the ‘thought’ and ‘low-level instruction’ largely duplicate each other with no reflection or high quality reasoning. By contrast, Appendix F of our paper shows our CoT that includes a detailed task plan, state-change understanding, error identification, and correction.
>
> (2) UI-Tars states it adds reflection in Section 4.4, but it does not describe how these reflections are constructed, nor provide data-quality or scale details. We think reflection and state-transition understanding are essential for agents, so we attach them to every step of action.
>
> (3) Evidence suggests our CoT is more helpful for task completion:
> - Aguvis evaluates OSWorld with L1 CoT and observes that L2 CoT actually lowers success (10.3 → 8.1).
> - UI-TARS (Section 5.5, Fig. 8) shows that adding thought reduces Pass@1 performance; only Pass@n improves.
> - In our ablation study (Section 5 Table 5), L2 (thought + action) outperforms L1 on Pass@1 for OSWorld, demonstrating the effectiveness of our richer CoT.
> ## Response to Weakness2:
> We sincerely thank the reviewer for asking these useful details.
>
> (1) Task selection: We first surveyed the most popular websites and applications across a wide range of domains—entertainment, office tools, and more. We selected the 200 + most widely used ones. Tasks were not pre-assigned, because annotators’ familiarity varies (especially with professional tools). We set a limit number for the apps and allowed annotators to choose. Annotators could also brainstorm new tasks with provided relevant YouTube tutorials so they could explore and create additional tasks.
>
> (2) Annotation schedule and cost: Annotating 22K tasks takes 6 months. All annotators are part-time. The total annotation cost was about USD 20,000. Annotation speed is roughly ten tasks per hour. The cost of synthesizing CoT costs USD 0.6 per task on average. The total cost of building this dataset is about USD 32,000.
>
> (3) Action distribution: The following tables summarize action frequencies across systems. Up to now, Ubuntu has 23,797 tasks, Windows 12,431, and macOS 5,200. Click is the dominant action—more than 60% on three systems. Hardware and usage patterns drive the secondary behaviors: macOS trackpads lead to heavier vertical/horizontal scrolling and more hotkey use; Windows mouse workflows show higher proportions of right-click and middle-click; and Ubuntu’s keyboard-centric, terminal-oriented culture results in the greatest shares of text input.
>
> | System | total_tasks | total_steps | total_actions |
> | --- | --- | --- | --- |
> | Windows | 12,431 | 239,767 | 280,174 |
> | macOS | 5,200 | 99,337 | 118,018 |
> | Ubuntu | 23,797 | 386,328 | 401,477 |
>
> | System | click | doubleClick | dragTo | hotkey | hscroll | middleClick | moveTo | press | rightClick | scroll | terminate | tripleClick | write |
> | --- | --- | --- | --- | --- | --- | --- | --- | --- | --- | --- | --- | --- | --- |
> | Windows | 60.03 | 1.99 | 4.35 | 1.52 | 0.00 | 0.01 | 8.95 | 5.27 | 1.10 | 4.51 | 4.42 | 0.00 | 7.85 |
> | macOS | 57.22 | 2.20 | 3.48 | 2.48 | 0.46 | 0.00 | 9.02 | 6.39 | 1.01 | 5.52 | 4.40 | 0.08 | 7.73 |
> | Ubuntu | 63.62 | 3.19 | 2.30 | 2.21 | 0.00 | 0.00 | 4.53 | 3.78 | 2.40 | 2.06 | 5.89 | 0.22 | 9.81 |
> ## Response to Weakness 3:
> We sincerely apologize that, due to time and space constraints of the submission deadline, we omitted details of the reflective long-form CoT synthesis pipeline from Appendix Section 5.
>
> Our CoT synthesis framework extends the pipelines of Aguvis [1]. We enrich the 'Thought' with more comprehensive agent components—especially state-transition perception and reflection. As illustrated in Fig. 13, the pipeline comprises three modules: reflector, generator, and summarizer.
> - Reflector. For each step it (i) compares pre- and post-action screenshots, (ii) checks whether the action code matches the UI, and (iii) verifies that the generated CoT is consistent. If a step is incorrect or redundant, the reflector explains why and the step is excluded from training. Otherwise, it describes the state changes the action produces.
> - Generator. Conditioned on the full agent context—prior reflections, action history, task goal, screenshots, and action code—it produces structured CoT. To improve grounding of coordinate-based actions, we add visual cues: a red marker at the click coordinate and a zoomed-in image patch.
> - Summarizer. It rewrites vague user-supplied goals into precise, aligned objectives and scores each trajectory for alignment, efficiency, and difficulty.
>
> All three modules are instantiated with claude-3-7-sonnet-20250219. Reflection enables the agent to spot earlier errors, adjust future plans, and return the task to the correct track; Appendix F shows an example of error identification and correction. Ablations in Section 5, Table 4, show that this module is an important driver of the performance gains.
> ## Response to Weakness 4:
> We thank the reviewer for suggesting an error analysis of the live OSWorld benchmark. After inspecting our online evaluation logs, we group the failures into the following categories:
>
> 1. Insufficient task knowledge: Foundation models still lack domain-specific GUI knowledge. Each application has unique UI conventions and operation logic; without explicit exposure, both agents and even humans can struggle.
>    Example: In the task “I have a lookup table for the officers of each branch. Please fill the second table using VLOOKUP.” the agent does not know the VLOOKUP function and therefore fails.
>
> 2. High-precision grounding errors: Tasks that require pixel-accurate actions frequently fail.
>    Example: In “Change the 2 in ‘H₂O’ to a subscript,” the agent must precisely drag-select only the “2,” but it often selects more characters.
>
> 3. Action repetition and failure to terminate: When the agent misjudges a state after an action, it may keep executing (and reflecting on) the same incorrect action. Occasionally it realizes and does a different action to break the loop, but sometimes it loops indefinitely. In some cases, the agent fails to notice that the task is already complete and keeps acting instead of terminating.
>
> 4. Long horizon tasks: OSWorld contains tasks requiring > 30-50 gold actions. Maintaining context over so many steps remains challenging.
>    Example: “Organize my desktop by placing academic papers in ‘Paper_reading’, coding projects in ‘Projects’, and everything else in ‘Miscellaneous’. For files without clear names, determine the category by content.”
>
> 5. Error perception: Although the agent model has a certain level of ability to identify errors and reflect, in some cases that requires high accuracy, it is still challenging. In fine-grained text-editing tasks, the agent may insert a word one character off, yet judge the action correct and proceed.
>
> We compared the tasks in the Pass@n experiment and find that the main difference between different runs comes from these factors:
>
> 1. The agent chooses different solutions in different runs. For example, in the task “Re-open the last closed browser tab,” the agent sometimes uses Ctrl + Shift + T (only one step) and other times navigates through the history menu (needs many steps). Agents may fail on harder solutions.
>
> 2. Minor omissions or extras. In Chrome or VS Code settings, forgetting to click “Save” (or performing an additional stray click) converts a correct solution into failure.
>
> 3. Environment dynamics: Occasional CAPTCHA dialogs, machine variability, and network latency can change the interaction sequence and lead to inconsistent outcomes.
>
> 4. Tasks that consistently fail. Most persistent failures stem from either insufficient task knowledge or long-horizon complexity. In some cases, the agent has no idea how to begin the task at all; in others, the task requires numerous delicate, precision actions, and the agent always breaks down in the middle of the task.
> ## Response to Limitation:
> We apologize that due to the limit of the paper length, we have written our Limitations in the Appendix Section A in our submission.
> ## Latest scaling results:
> Over the past two months we further expanded both data and models: the dataset now contains 40 K trajectories with additional grounding data, and we fine-tuned Qwen 2.5-VL-7B and Qwen 2.5-VL-32B on our dataset. The 7B model reaches a 27.9 % success rate, and the 32 B model averages **34.8 %**, which is the state-of-the-art for open-source agent models and surpasses GPT-4o-based OpenAI CUA.
>
> | Model | OSWorld |
> | --- | --- |
> | UI-TARS-72B-DPO| 27.1 %  |
> | OpenAI CUA (GPT-4o) | 31.4 % |
> | AgentNet-Qwen2.5-VL-7B | 27.9 %  |
> | AgentNet-Qwen2.5-VL-32B | **34.8 %**  |
>
> [1] Xu, Yiheng, Zekun Wang, Junli Wang, Dunjie Lu, Tianbao Xie, Amrita Saha, Doyen Sahoo, Tao Yu, and Caiming Xiong. "Aguvis: Unified pure vision agents for autonomous gui interaction." arXiv preprint arXiv:2412.04454 (2024).
>
> [2] Qin, Yujia, Yining Ye, Junjie Fang, Haoming Wang, Shihao Liang, Shizuo Tian, Junda Zhang et al. "Ui-tars: Pioneering automated gui interaction with native agents." arXiv preprint arXiv:2501.12326 (2025).

---

> > ### Comment · Area_Chair_pBRN · 2025-08-04
> > **Please respond to authors**
> >
> > Dear reviewer,
> >
> > Now that the authors have provided their rebuttal, do you have any additional thoughts?
> >
> > Thanks,
> > AC

---

> > ### Comment · Reviewer_6mQE · 2025-08-04
> >
> > Thanks the authors for the detailed response. I find the data statistics and error analysis provide a lot of useful insights into the dataset and the agent. The SOTA results further prove the usefulness of the dataset, which did take a lot of efforts to create based on the cost details provided by the authors. I'd suggest adding these details in the rebuttal into the paper, and try to include more content if possible for 1) differentiating from previous approaches, 2) more error / result analysis for future directions. I've increased my score accordingly.

---

> > > ### Author Response · Authors · 2025-08-05
> > >
> > > We sincerely thank Reviewer 6mQE for the follow-up suggestions and for considering an increased score. The questions and detailed feedback provided in the rebuttal have been extremely helpful for improving AgentNet. We will incorporate all your points in our next revision, especially the distinctions between our method and others, as well as the error analysis section. Thank you again for your constructive and insightful engagement!

---

### Decision · Program_Chairs · 2025-09-17

**Decision:**

Accept (spotlight)

**Comment:**

This paper introduces AgentNet, a comprehensive open-source framework for developing, training, and evaluating computer-use agents (CUAs). The work aims to bridge the gap between proprietary and open-source CUA models by providing essential infrastructure, data, and benchmarks to the research community. The reviewers appreciate the design and effectiveness (as shown from the experiments) of the proposed pipeline as well as the thorough evaluation and analysis.

The reviewers identified several concerns, such as limited novelty, performance (the proposed architectures while outperforming some baselines, did not outperform larger models, limited error analysis, and others. The authors, for the most part, addressed the concerns satisfactorily which the reviewers appreciated. While only one reviewer increased their score, all reviewers are very positive about this work.